# Pretrained Optimization Model for Zero-Shot Black Box Optimization

**Xiaobin Li**
Xidian University
22171214784@stu.xidian.edu.cn

**Kai Wu**∗
Xidian University
kwu@xidian.edu.cn

**Yujian Betterrest Li**
Xidian University
bebetterest@outlook.com

**Xiaoyu Zhang**
Xidian University
xiaoyuzhang@xidian.edu.cn

**Handing Wang**
Xidian University
hdwang@xidian.edu.cn

**Jing Liu**
Xidian University
neouma@mail.xidian.edu.cn

## Abstract

Zero-shot optimization involves optimizing a target task that was not seen during training, aiming to provide the optimal solution without or with minimal adjustments to the optimizer. It is crucial to ensure reliable and robust performance in various applications. Current optimizers often struggle with zero-shot optimization and require intricate hyperparameter tuning to adapt to new tasks. To address this, we propose a Pretrained Optimization Model (POM) that leverages knowledge gained from optimizing diverse tasks, offering efficient solutions to zero-shot optimization through direct application or fine-tuning with few-shot samples. Evaluation on the BBOB benchmark and two robot control tasks demonstrates that POM outperforms state-of-the-art black-box optimization methods, especially for high-dimensional tasks. Fine-tuning POM with a small number of samples and budget yields significant performance improvements. Moreover, POM demonstrates robust generalization across diverse task distributions, dimensions, population sizes, and optimization horizons. For code implementation, see https://github.com/ninja-wm/POM/.

## 1 Introduction

Black box optimization, including tasks like hyperparameter optimization (HPO) [1], neuroevolution [2–4], neural architecture search (NAS) [5], and algorithm selection [6], is very important. In these scenarios, the algorithm can evaluate $f(\mathbf{x})$ for any solution $\mathbf{x}$; however, access to additional information about $f$, such as the Hessian and gradients, is unavailable.

Addressing diverse BBO problems necessitates the tailored design of specific algorithms to achieve satisfactory performance. Crafting these algorithms typically demands substantial expertise. Therefore, it is crucial to ensure reliable and robust performance of the optimizer in various applications, called zero-shot optimization. Zero-shot optimization involves optimizing a target task that was not seen during training, aiming to provide the optimal solution without or with minimal adjustments to the optimizer.

---

∗Corresponding author

The studies [7–9] employed Transformer or diffusion models to pretrain model-based optimizers using offline datasets. While effective, these methods primarily fit optimization trajectories of other BBO algorithms to a specific task, potentially requiring retraining for new tasks, limiting their ability to zero-shot optimization. Subsequently, [10, 11] introduced two learned optimization frameworks for meta-learning evolution strategy (ES) and genetic algorithm (GA). However, the performance of these two methods on zero-shot optimization is weaker than that of CMA-ES [12] (see Section 4.2).

To address zero-shot optimization, especially for continuous optimization, we introduce a population-based Pretrained Optimization Model, called POM. Leveraging multiple individuals, population-based optimizers gain a better understanding of the fitness landscape. The core of the optimizer is how to design optimization strategies that sample better solutions. Inspired by the solution-producing mechanism of evolutionary computation, we design powerful POM blocks to form a general optimization strategy representation framework. Drawing inspiration from [13], we introduce an end-to-end gradient-based training method for POM, termed *MetaGBT* (Meta Gradient-Based Training), ensuring stable and rapid training for POM. Pretraining POM on a set of training functions with *MetaGBT* ensures good optimization strategy. Our contributions can be summarized as follows:

- **Excellent ability to solve zero-shot BBO**. We develop a efficient POM for zero-shot BBO, demonstrating a substantial performance advantage over state-of-the-art black-box optimizers.
- **Excellent ability to solve few-shot BBO**. Few-shot optimization is the existence of a small budget of function evaluations for the target task to tune the optimizer for better performance. More than 30% performance improvement can be obtained with 25 random function evaluations.

## 2 Related Work

**Heuristic Population-based BBO Algorithms**. Numerous metaheuristic population-based algorithms, such as genetic algorithms [14], evolution strategies [15–17], particle swarm optimization [18, 19], and differential evolution [20, 21], have been devised to address optimization problems. Notably, CMA-ES [12] and L-SHADE [22] stand out as state-of-the-art methods for BBO. However, these approaches rely on manually designed components, exhibiting inefficiency and fragility when confronted with new tasks. In contrast, the proposed POM can autonomously acquire optimization strategies from problem instances, mitigating the aforementioned limitations.

**Pretrained Population-based BBO Algorithms**. Pre-training BBO algorithms can be categorized into two types within the meta-learning framework. The first type frames meta-learning BBO algorithms as a bi-level optimization problem [23]. For instance, [24] leverages meta-learning to infer population-based black-box optimizers that automatically adapt to specific task classes. LES [25] designs a self-attention-based search strategy for discovering effective update rules for evolution strategies through meta-learning. Subsequent works like LGA [10] utilize this framework to discover the update rules of Gaussian genetic algorithms via Open-ES [26]. The second type models the meta-learning of a BBO algorithm as a reinforcement learning problem. [27] meta-learn a policy that adjusts the mutation step-size parameters of CMA-ES [12]. Category one faces the curse of dimensionality, where an escalating number of model parameters leads to skyrocketing training difficulty, impeding the development of intricate strategies. In contrast, category two, which models meta-learning optimizers as reinforcement learning tasks, grapples with training instability. POM, employing a gradient-based end-to-end training approach, successfully bypasses the curse of dimensionality, ensuring stable training.

**LLM for Optimization**. In line with POMs, various optimization approaches leveraging Large Language Models (LLMs) have emerged to address diverse problem domains, including NP-hard problems [28, 29], algorithm evolution [30–33], reward design [34], and Neural Architecture Search (NAS) [35, 36]. Notably, LLMs play a role in sampling new solutions. However, their optimization strategies depend on externally introduced natural selection mechanisms and are less effective in numerical optimization scenarios [37]. LLaMoCo [38] and EoH [39] use LLM to generate code to solve optimization problems, but the performance of LLaMoCo depends on carefully designed instructions and prompts, and EoH has expensive evaluation costs. TNPs [40], ExPT [41] and LICO [42] use transformer structures to solve the BBO problem and have achieved good results. However, TNPs requires contextual information of the target problem, and neither ExPT nor LICO can be

directly used to solve tasks with different dimensions from the training task. These methods lack the universal applicability as pretrained BBO models due to a deficiency in generating capabilities across tasks.

All the above methods cannot be the zero-shot optimizer. The first two categories need to adjust the hyperParameters when optimizing the new tasks, while the latter must fine-tune the instructions to achieve satisfactory results.

## 3 Pretrained Optimization Model

### 3.1 Problem Definition

A black-box optimization problem can be transformed as a minimization problem, and constraints may exist for corresponding solutions: $\min_{\mathbf{x}} f(\mathbf{x}), s.t. \ x_i \in [l_i, u_i]$, where $\mathbf{x} = (x_1, x_2, \cdots, x_d)$ represents the solution of optimization problem $f$, the lower and upper bounds $\mathbf{l} = (l_1, l_2, \cdots, l_d)$ and $\mathbf{u} = (u_1, u_2, \cdots, u_d)$, and $d$ is the dimension of $\mathbf{x}$. For more background information on evolutionary algorithms, see Appendix A.

**Definition 1 Zero-shot Optimization**. *Zero-shot optimization refers to an optimizer that is applied directly to solve a continuous black-box optimization problem $f$ without any tuning. This means that the optimizer does not require any contextual information about $f$ and can be directly used to handle problems of any dimensionality.*

**Definition 2 Few-shot Optimization**. *Alternatively, it is permissible to fine-tune the optimizer using a small portion of the function evaluation budget for the objective task, and then use the fine-tuned optimizer to solve $f$.*

### 3.2 Classic Population Optimization Algorithm

In this section, we use Differential Evolution (DE) as an example to review classic evolutionary algorithms. DE [20, 43] is a prominent family within evolutionary algorithms (EAs), known for its advantageous properties such as rapid convergence and robust performance [44, 45]. The optimization strategy of DE primarily involves mutation and crossover operations.

The classic DE/rand/1 crossover operator is illustrated in Eq. (1) (additional examples are listed in Appendix A.2). Each mutation strategy can be viewed as a specific instance of Eq. (2); Further details are provided in Appendix A.2. Additionally, we represent the mutation strategy in a matrix form, as shown in Eq. (3). The matrix $\mathbf{S}$ evolves with the generation index $t$, indicating that the mutation strategy adapts across different generations. Consequently, we propose a module to enhance the performance of the mutation operation, which leverages the information from the population of the $t$th generation to generate $\mathbf{S}^t$. This serves as the motivation for our design of the LMM.

$$\mathbf{v}_i^t = \mathbf{x}_{r1}^t + F \cdot (\mathbf{x}_{r2}^t - \mathbf{x}_{r3}^t) \tag{1}$$

In the crossover phase at step $t$, DE uses a fixed crossover probability $cr_i^t \in [0, 1]$ for each individual $\mathbf{x}_i^t$ in the population, as shown in Eq. (9). The crossover strategy for the entire population can then be expressed as a vector $\mathbf{cr}^t = (cr_1^t, cr_2^t, \cdots, cr_N^t)$. Our goal is to design a module that adaptively generates $\mathbf{cr}^t$ using the information from the population. This approach allows for the automatic design of the crossover strategy by controlling the parameter $cr$. This serves as the motivation for our design of LCM.

### 3.3 Design of POM

A population consists of $n$ individuals, denoted as $\mathbf{X} = \{\mathbf{x}_1, \mathbf{x}_2, \cdots, \mathbf{x}_n\}$. In this paper, $\mathbf{X}$ is also treated as $\mathbf{X} = [\mathbf{x}_1, \mathbf{x}_2, \cdots, \mathbf{x}_n]^T$ to support matrix operations. We feed POM an initial random population $\mathbf{X}^0$ at step 0, specify the evolution generation $T$ for it, and hope that it can generate a population $\mathbf{X}^T$ close to the global optimum at step $T$, as shown in $\mathbf{X}^T = POM(\mathbf{X}^0, T|\theta)$, where $\theta \in \Omega$ is the parameters of POM, where $\Omega$ stands for the strategy space. The goal of training POM is to find an optimal $\theta$ in $\Omega$. As shown in Fig. 1, POM consists of LMM, LCM and SM.

**LMM** LMM generates candidate solutions $\mathbf{v}_i^t$ for individual $\mathbf{x}_i^t$ through Eq. (2), which enables the population information to be fully utilized in the process of generating candidate solutions $\mathbf{v}_i^t$.

$$\mathbf{v}_i^t = \sum_j^N w_{i,j} \mathbf{x}_j^t \quad (\forall w_{i,j} \in \mathbb{R}, w_{i,i} \neq 0) \tag{2}$$

Further, we organize Eq. (2) into a matrix form, as shown in Eq. (3).

$$\mathbf{V}^t = \mathbf{S}^t \times \mathbf{X}^t \tag{3}$$

$\mathbf{X}^t \in \mathbb{R}^{N \times d}$ is the population in generation $t$ and $\mathbf{S}^t \in \mathbb{R}^{N \times N}$. $\mathbf{S}$ evolves with each change in $t$, signifying a mutation strategy that adapts across generations. Consequently, it is imperative to devise a module that leverages information from the population at generation $t$ to generate $\mathbf{S}^t$. Any mutation operator of differential evolution, such as the classic DE/rand/1 mutation operator, can be converted into Equation (3) in the specific case of $\mathbf{S}$ (see Appendix A.2 for details). At the same time, the crossover operation of GAs can also be generalized into the form of Equation (3) [46].

The function of LMM is designed based on Multi-head self-attention (MSA) [47], as shown as follows:

$$\mathbf{S}^t = LMM(\mathbf{H}^t | \theta_1) \tag{4}$$

where $\theta_1 = \{\mathbf{W}_{m1}, \mathbf{W}_{m2}, \mathbf{W}_{m3}, \mathbf{b}_{m1}, \mathbf{b}_{m2}, \mathbf{b}_{m3}\}$ denotes the trainable parameters within LMM, while $\mathbf{H}^t = [\mathbf{h}_1^t, \mathbf{h}_2^t, \cdots, \mathbf{h}_N^t]$ serves as LMM's input, encapsulating population information. Each $\mathbf{h}_i^t$ incorporates details about $\mathbf{x}_i^t$, encompassing: 1) $\hat{f}_i^t$: the normalized fitness $f(\mathbf{x}_i^t)$ of $\mathbf{x}_i^t$; 2) $\hat{r}_i^t$: the centralized ranking of $\mathbf{x}_i^t$. The method for calculating $\hat{f}_i^t$ is:

$$\hat{f}_i^t = \frac{f(\mathbf{x}_i^t) - \mu^t}{\sigma^t} \tag{5}$$

where $\mu^t$ and $\sigma^t$ denote the mean and standard deviation, respectively, of individual fitness values within the population at time $t$. We build $\hat{r}_i^t$ as follows:

$$\hat{r}_i^t = \left(\frac{rank(\mathbf{x}_i^t, \mathbf{X}^t)}{N} - 0.5\right) \times 2 \tag{6}$$

where *rank* yields the ranking of $\mathbf{x}_i^t$ within the population $\mathbf{X}^t$, with values ranging from 1 to $N$. Thus, LMM utilizes information on the relative fitness of individuals to dynamically generate the strategy $\hat{\mathbf{S}}^t$. $\hat{r}_i^t$ serves as position encoding, explicitly offering the ranking information of individuals. Equation (7) details the computation of $\hat{\mathbf{S}}^t$.

$$\hat{\mathbf{H}}^t = Tanh(\mathbf{H}^t \times \mathbf{W}_{m1} + \mathbf{b}_{m1}), \quad \mathbf{Q}^t = Tanh(\hat{\mathbf{H}}^t \times \mathbf{W}_{m2} + \mathbf{b}_{m2})$$
$$\mathbf{K}^t = Tanh(\hat{\mathbf{H}}^t \times \mathbf{W}_{m3} + \mathbf{b}_{m3}), \quad \hat{\mathbf{S}}^t = Tanh\left(\frac{\mathbf{Q}^t \times (\mathbf{K}^t)^T}{\sqrt{(d_m)}}\right) \tag{7}$$

where *Tanh* is an activation function. $\mathbf{W}_{m1} \in \mathbb{R}^{2 \times d_m}$ and $\mathbf{W}_{m2}, \mathbf{W}_{m3} \in \mathbb{R}^{d_m \times d_m}$. $\mathbf{b}_{m1}, \mathbf{b}_{m2}$, and $\mathbf{b}_{m3}$ are vector with dimension $d_m$. $\hat{\mathbf{H}}^t \in \mathbb{R}^{N \times d_m}$, $\mathbf{Q}^t, \mathbf{K}^t \in \mathbb{R}^{N \times d_m}$, and $\hat{\mathbf{S}}^t \in \mathbb{R}^{N \times N}$.

The topological structure of the population significantly influences their information exchange [48]. When all individuals engage in information exchange, the algorithm's convergence may suffer, diversity could diminish, and susceptibility to local optima increases. To address this, we introduce a *mask* operation during both training and testing phases, where the probability of setting each element in $\hat{\mathbf{S}}^t$ to 0 is $r_{mask}$. This operation enhances POM's ability to learn efficient and robust strategies, as validated in our experiments. Consequently, $\mathbf{S}^t$ is derived using Eq. (8).

$$\mathbf{S}^t = mask(\hat{\mathbf{S}}^t | r_{mask}) \tag{8}$$

Finally, we get $\mathbf{V}^t$ via Eq. (3).

**LCM** For each individual $\mathbf{x}_i^t$ at step $t$, a crossover probability $cr_i^t \in [0, 1]$ is established. Consequently, the population's crossover strategy is encapsulated in the vector $\mathbf{cr}^t = (cr_1^t, cr_2^t, \cdots, cr_N^t)$. The crossover operation, as depicted in Eq. (9), can be elucidated as follows:

$$\mathbf{u}_{i,k}^t = \begin{cases} \mathbf{v}_{i,k}^t, & \text{if} \quad rand(0,1) \leq cr_i^t \\ \mathbf{x}_{i,k}^t, & \text{otherwise} \end{cases} \quad \forall i \in [1, N] \tag{9}$$

The module design should facilitate the adaptive generation of $\mathbf{cr}^t$ by leveraging population information. Executing the crossover operation with $\mathbf{cr}^t$ yields $\mathbf{U}^t = [\mathbf{u}_1^t, \mathbf{u}_2^t, \cdots, \mathbf{u}_N^t]$.

LCM is designed based on FFN [47], as shown in Eq. (10),

$$\mathbf{cr}^t = LCM(\mathbf{Z}^t | \theta_2) \tag{10}$$

where $\theta_2 = \{\mathbf{W}_{c1}, \mathbf{b}_{c1}, \mathbf{W}_{c2}, \mathbf{b}_{c2}, \tau\}$ is the parameter of LCM and $\mathbf{Z}^t \in \mathbb{R}^{N \times 3}$ is the population information used by LCM. Here, $\mathbf{Z}^t = [\mathbf{z}_1^t, \mathbf{z}_2^t, \cdots, \mathbf{z}_N^t]$. $\mathbf{z}_i^t$ represents the relevant information of individual $\mathbf{x}_i^t$ and $\mathbf{X}^t$. For example, it can include the ranking information of $\mathbf{x}_i^t$, the fitness information of $\mathbf{x}_i^t$, the Euclidean distance between $\mathbf{x}_i^t$ and $\mathbf{V}_i^t$, and the distribution information of individuals within the population (such as the fitness distribution, the distance between pairs of individuals), etc. In this paper, $\mathbf{z}_i^t$ includes the following information as a case study: 1) $\hat{f}_i^t$: the normalized fitness $f(\mathbf{x}_i^t)$ of $\mathbf{x}_i^t$; 2) $\hat{r}_i^t$: the centralized ranking of $\mathbf{x}_i^t$; 3) $sim_i^t$: the cosine similarity between $x_i^t$ and $v_i^t$.

$$\mathbf{h}^t = Tanh(\mathbf{Z}^t \times \mathbf{W_{c1}} + \mathbf{b_{c1}}), \quad \hat{\mathbf{h}}^t = layernorm(\mathbf{h}^t | \tau), \quad \mathbf{cr}^t = Sigmoid(\hat{\mathbf{h}}^t \times \mathbf{W_{c2}} + \mathbf{b_{c2}}) \tag{11}$$

where the activation function *Sigmoid* maps inputs to the range $(0, 1)$. $\mathbf{W_{c1}} \in \mathbb{R}^{3 \times d_c}$, $\mathbf{W_{c2}} \in \mathbb{R}^{d_c \times 1}$, $\tau$ is the learnable parameters of *layernorm* [49]. $\mathbf{b_{c1}}$ and $\mathbf{b_{c2}}$ are vectors with dimensions $d_c$ and 1, respectively.

Although we derive $\mathbf{cr}^t$ from Eq. (11) as in Eq. (9), the discrete nature of the crossover operator renders it non-differentiable, impeding gradient-based training of the *LCM* module. To address this limitation, we introduce the *gumbel_softmax* method [50], providing an efficient gradient estimator that replaces non-differentiable samples from a categorical distribution with differentiable samples from a novel Gumbel-Softmax distribution.

Eq. (12) shows how to perform crossover operations between $\mathbf{x}_i^t$ and $\mathbf{v}_i^t$ in *LCM* ($\forall i \in [1, N]$).

$$\begin{aligned}
\mathbf{r}_i^t &= rand(d), \quad \mathbf{cv}_i^t = gumbel\_softmax(cat(\mathbf{r}_i^t, tile(cr_i^t, d))), \\
\mathbf{u}_i^t &= \mathbf{cv}_{i,0}^t \cdot \mathbf{x}_i^t + \mathbf{cv}_{i,1}^t \cdot \mathbf{v}_i^t, \quad \mathbf{U}^t = [\mathbf{u}_1^t, \mathbf{u}_2^t, \cdots, \mathbf{u}_N^t]
\end{aligned} \tag{12}$$

First, the *rand* function samples uniformly from the range $[0, 1]$ to obtain a vector $\mathbf{r}_i^t$. Then get $cr_i^t$ from $\mathbf{cr}^t$ according to the index. The *tile* function expands $cr_i^t$ into a $d$-dimensional vector: $[cr_i^t, cr_i^t, \cdots, cr_i^t]$. The *cat* function concatenates them into a matrix as shown below:

$$\begin{bmatrix} r_{i,1}^t & r_{i,2}^t & \cdots & r_{i,d}^t \\ cr_i^t & cr_i^t & \cdots & cr_i^t \end{bmatrix} \tag{13}$$

Here, *gumbel_softmax* is executed column-wise. For any column, the larger element becomes 1 after *gumbel_softmax* and 0 otherwise. Therefore, $\mathbf{cv}_i^t \in \mathbb{R}^{2 \times d}$ may be a matrix like this:

$$\mathbf{cv}_i^t = \begin{bmatrix} 1 & 0 & 0 & 0 & 1 & 1 & \cdots & 1 & 1 \\ 0 & 1 & 1 & 1 & 0 & 0 & \cdots & 0 & 0 \end{bmatrix} \tag{14}$$

(a) POM      (b) training      (c) testing

Figure 1: In the figure, $\mathbf{X}^0$ is the initial random population. (a) The overall architecture of the POM. (b) POM training process. Here $T$ is the size of the inner loop iteration step during training, and the training function should be differentiable. (c) POM testing process. Here, $T$ is the number of iterations of the testing process and $f$ is the target task. $f$ does not have to be differentiable. Here we directly apply the trained POM to solve $f$ without requiring gradient information.

**Overall Framework** We design LMM and LCM to achieve the generation of sample strategy (that is, generate $\mathbf{S}^t$) and crossover strategy (that is, generate $\mathbf{cr}^t$), respectively. The overall architecture of POM is shown in Fig. 1. The parameters that need to be trained in *POM* are $\theta = \{\theta_1, \theta_2\}$. At time step $t$, the population is $\mathbf{X}^t$. Initially, we amalgamate the information from $\mathbf{X}^t$ to construct descriptive representations of the population, $\mathbf{H}^t$ and $\mathbf{Z}^t$. *LMM* adaptively generates $\mathbf{S}^t$ based on $\mathbf{H}^t$.

The multiplication of $\mathbf{X}^t$ and $\mathbf{S}^t$ yields $\mathbf{V}^t$ (see Eq. (3)). Next, *LCM* adaptively generates $\mathbf{cr}^t$ based on its input $\mathbf{Z}^t$, and performs a crossover operation based on $\mathbf{cr}^t$ to obtain $\mathbf{U}^t$. Finally, *SM* [51], a *1-to-1* selection strategy is executed between $\mathbf{U}^t$ and $\mathbf{X}^t$ to produce the next-generation population $\mathbf{X}^{t+1}$.

$$\mathbf{X}^{t+1} = SM(\mathbf{X}^t, \mathbf{U}^t) = tile(l_{x>0}(\mathbf{M}_{F'} - \mathbf{M}_F)) \odot \mathbf{X}^t + tile(1 - l_{x>0}(\mathbf{M}_{F'} - \mathbf{M}_F)) \odot \mathbf{U}^t \quad (15)$$

where $l_{x>0}(x) = 1$ if $x > 0$ and $l_{x>0}(x) = 0$ if $x < 0$, and the *tile* copy function extends the indication matrix to a tensor with size $(N, d)$, $\mathbf{M}_F(\mathbf{M}_{F'})$ denotes the fitness matrix of $\mathbf{X}^t(\mathbf{U}^t)$, and $\odot$ indicates the pairwise multiplication between inputs.

---

| **Algorithm 1** MetaGBT | **Algorithm 2** Driving POM to Solve Problem |
|---|---|
| **Input:** $T$, $n$, training set $TS$. | **Input:** Generations $T$, population size $n$, BBO problem $f$. |
| **Output:** The optimal $\theta$. | **Output:** The optimal $\mathbf{X}^T$ found. |
| 1: Randomly sample the parameter $\theta$ of *POM*. | 1: *POM* loads the trained parameter $\theta$. |
| 2: **while** not done **do** | 2: Randomly sample an initial population $\mathbf{X}^0$ of size $n$. |
| 3:     Sample $|TS|$ populations of size $n$ to obtain $[\mathbf{X}_1^0, \mathbf{X}_2^0, \cdots, \mathbf{X}_{|TS|}^0]$. | 3: **for** $t = 0, 1, \ldots, T - 1$ **do** |
| 4:     **for** $i = 1, 2, \ldots, |TS|$ **do** | 4:     Construct $\mathbf{H}^t$ based on $\mathbf{X}^t$ and $f$. |
| 5:         Randomly sample $\omega^i$ for the $f_i$ in $TS$. | 5:     $\mathbf{S}^t \leftarrow LMM(\mathbf{H}^t | \theta_1)$. |
| 6:     **end for** | 6:     $\mathbf{V}^t \leftarrow \mathbf{S}^t \times \mathbf{X}^t$. |
| 7:     **for** $t = 1, 2, \ldots, T$ **do** | 7:     Build $\mathbf{Z}^t$ based on $\mathbf{X}^t$, $\mathbf{V}^t$ and $f$. |
| 8:         **for** $i = 1, 2, \ldots, |TS|$ **do** | 8:     $\mathbf{cr}^t \leftarrow LCM(\mathbf{Z}^t | \theta_2)$. |
| 9:             $\mathbf{X}_i^t \leftarrow POM(\mathbf{X}_i^{t-1}, 1 | \theta)$. | 9:     Construct $\mathbf{U}^t$ using Equation (12). |
| 10:           $loss_i^t \leftarrow l_i(\mathbf{X}_i^t, \mathbf{X}_i^{t-1}, f_i, \omega^i, \lambda)$. | 10:     $\mathbf{X}^{t+1} \leftarrow SM(\mathbf{X}^t, \mathbf{U}^t)$. |
| 11:         **end for** | 11: **end for** |
| 12:     $\theta \leftarrow$ Update $\theta$ based on $\frac{1}{|TS|} \sum_i loss_i^t$. | |
| 13:     **end for** | |
| 14: **end while** | |

---

### 3.4 Tasks, Loss Function & MetaGBT

POM is meticulously crafted as a model amenable to end-to-end training based on gradients. While POM necessitates gradient information from the training task during the training phase, it exhibits the ability to tackle BBO problems in the testing phase without relying on any gradient information. To ensure the acquisition of an efficient, highly robust, and broadly generalizable optimization strategy, POM undergoes training on a diverse set of tasks. Training on these tasks sequentially poses the risk of domain overfitting, local optima entrapment, and diminished generalization performance. Consequently, we introduce a training methodology named *MetaGBT*.

**Tasks**. We form a training task set $TS = \{f_i(\mathbf{X} | \omega^j)\}$, where $i \in [1, 5]$ and $j \in [1, N]$, comprising $4N$ tasks derived from Table 3 in appendix, where $\omega_i$ denotes the task parameter influencing the function's landscape offset. Our selection of these functions for the training task is motivated by their diverse landscape features. The specific landscape features encompassed in $TS$ are detailed in Appendix B.

**Loss Function**. To avoid bias of different output scales in *TS*, for any function $f_i$ in $TS$, we design the normalized loss function $l_i(\mathbf{X}^t, \mathbf{X}^{t-1}, f_i, \omega^i, \lambda)$. In Equation (16), $l_i^1$ calculates the average fitness difference between the input and output of the POM, further normalized within $[0, 1]$. This encourages convergence of the algorithm. $l_i^2$ uses standard deviation to simulate the distribution of the output population, encouraging diversity in the output population. $std(\mathbf{X}^t, j)$ is the standard deviation of the jth dimension of the population. $\lambda$ is a hyperparameter, and we find that setting it to 0.005 can make model training more stable.

$$\mathbf{X}^t = POM(\mathbf{X}^{t-1}, 1 | \theta)$$

$$l_i^1 = \frac{\frac{1}{|\mathbf{X}^t|} \sum_{\mathbf{x} \in \mathbf{X}^t} f_i(\mathbf{x} | \omega^i) - \frac{1}{|\mathbf{X}^{t-1}|} \sum_{\mathbf{x} \in \mathbf{X}^{t-1}} f_i(\mathbf{x} | \omega^i)}{\left| \frac{1}{|\mathbf{X}^{t-1}|} \sum_{\mathbf{x} \in \mathbf{X}^{t-1}} f_i(\mathbf{x} | \omega^i) \right|}, \quad l_i^2 = \frac{\sum_{j=1}^d std(\mathbf{X}^t, j)}{d}, \quad l_i = l_i^1 - \lambda l_i^2 \quad (16)$$

**MetaGBT**. The pseudocode for *MetaGBT* is presented in Algorithm 1. Initially, we sample the *POM* parameter $\theta$ from a standard normal distribution. The objective of *MetaGBT* is to iteratively update $\theta$ to bring it closer to the global optimum $\theta^*$. In line 2, we sample a population for each task in *TS*. Lines 3, 4 and 5 involve the resampling of task parameters for all tasks in *TS*, thereby altering the task landscape, augmenting training complexity, and enhancing the learning of robust optimization strategies by POM. The final loss function (line 10) is determined by computing the average of the loss functions for all tasks. Subsequently, in line 12, we update $\theta$ using a gradient-based optimizer, such as Adam [52]. The trained *POM* is then ready for application in solving an unknown BBO problem, as depicted in Algorithm 1.

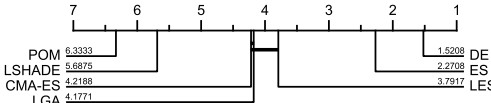

Figure 2: The critical difference diagram illustrates the performance ranking of seven algorithms across 24 BBOB problems with dimensions $d = 30, 100$, employing Wilcoxon-Holm analysis [53] at a significance level of $p = 0.05$. Algorithm positions are indicative of their mean scores across multiple datasets, with higher scores signifying a method consistently outperforming competitors. Thick horizontal lines denote scenarios where there is no statistically significant difference in algorithm performance.

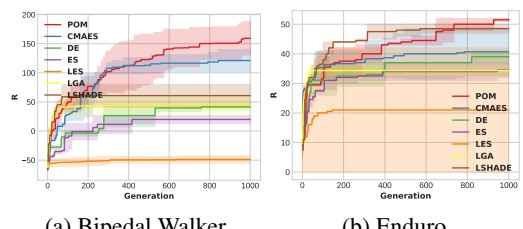

(a) Bipedal Walker      (b) Enduro

Figure 3: Experimental results are presented for the Bipedal Walker (a) and Enduro (b), with the vertical axis denoted as $R$, representing the strategy score. The score corresponds to the total reward acquired by the agent during interactions with the environment.

## 4 Experiments

### 4.1 Experimental Setup

We test the performance of POM on the widely used BBO benchmark and two complex real-world problems (see Appendix C). Selected methods include DE (DE/rand/1/bin) [54] and ES (($\mu,\lambda$)-ES) as population-based baselines, L-SHADE [22] and CMA-ES [12] as state-of-the-art population-based BBO methods, and LES [25] and LGA [10] as state-of-the-art POMs. POM is trained on $TS$ with $T = 100$, $n = 100$, and $d = 10$. Detailed parameters for all compared methods are provided in Appendix E. Please refer to Appendix D for the reasons for choosing these algorithms.

### 4.2 Results

**BBOB [55]**. We evaluate the generalization ability of POM across 24 BBOB functions with dimensions $d = 30$ and $d = 100$, where optimal solutions are located at **0**. Figure 2 presents the critical difference diagram comparing all algorithms (refer to Appendix Tables 4 and 6, and Figures 11, 12 and 13 for detailed results). POM significantly outperforms all methods, showcasing its efficacy across varying dimensions. Despite being trained solely on TF1-TF4 with $d = 10$, POM excels in higher dimensions ($d = \{30, 100, 500\}$), with its performance advantage becoming more pronounced with increasing dimensionality. Particularly on complex problems F21-F24, where global structure is weak, POM lags behind LSHADE but surpasses other methods, attributed to its adaptability through fine-tuning. TurBO [56] is the Bayesian optimization algorithm with the best performance on BBOB [57]. Under little budget conditions, the performance of POM outperforms that of TurBO in most cases (see Appendix G for details).

**Bipedal Walker [58]**. The Bipedal Walker task involves optimizing a fully connected neural network with $d = 874$ parameters over $k = 800$ time steps to enhance robot locomotion control. In Fig. 3(a), LSHADE shows ineffectiveness, while CMA-ES, LSHADE, and LGA suffer from premature convergence. Conversely, POM achieves stable and swift convergence, ultimately attaining the highest score.

**Enduro [58]**. Enduro task entails controlling a strategy with $d = 4149$ parameters across $k = 500$ steps, posing greater difficulty than Bipedal Walker. As depicted in Fig. 3(b), LGA and LES exhibit premature convergence and limited exploration. While CMA-ES initially converges slightly

faster than POM, the latter maintains a superior balance between exploration and exploitation, outperforming LSHADE.

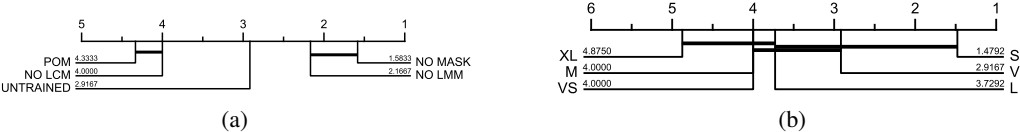

(a)                                            (b)

Figure 4: (a) Results of ablation study. The metric used to evaluate performance is the optimal value of the function found, with smaller values being better. Here, $d = 30$. (b) Results of POMs with different sizes on BBOB tests ($d = 100$).

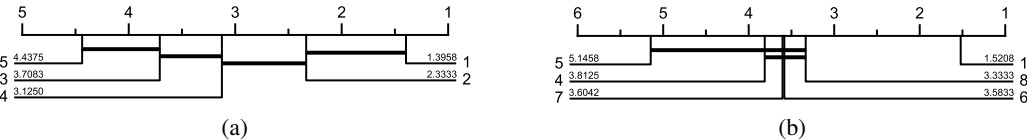

(a)                                            (b)

Figure 5: The impact of training dataset size on the performance of POM. $d = 100$. 1 means that the training set only contains $TF1$, and 2 means that the training set only contains $TF1$ and $TF2$, and so on.

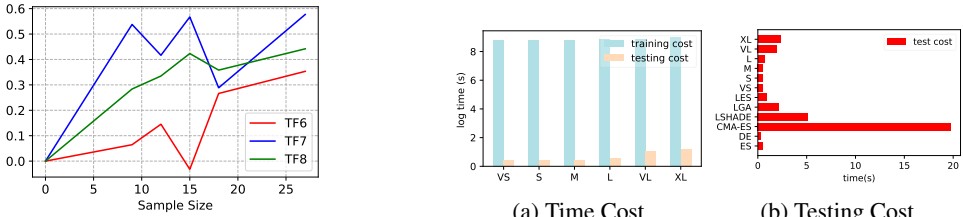

Figure 6: Experimental results of fine-tuning tests. $RFI = \frac{performance\ improvement}{performance\ of\ base\ POM}$.

(a) Time Cost          (b) Testing Cost

Figure 7: (a) Time cost of POM. (b) Testing cost of baselines and POM.

## 4.3 Analysis

**Ablation Study**    The ablation study results for the designed modules are presented in Fig. 4 (a) (refer to Appendix Table 7 for additional details). Configurations include *UNTRAINED*, representing an untrained POM with randomly initialized parameters; *NO LMM*, where the LMM is excluded, and a simple *DE/rand/1/bin* mutation operator is employed; *NO LCM*, indicating the absence of the learnable crossover operation, using only binomial crossover; and *NO MASK*, signifying the omission of the mask operation described in Eq. (8).

While *UNTRAINED* yields optimal results for F9 and F16, as an untrained POM is inherently an optimization strategy, the adaptability of trained POM surpasses the baselines in most scenarios. In simpler tasks with $d = 30$, *UNTRAINED* underperforms, demonstrating the advantage of trained POM on more complex tasks. Notably, *NO LMM* and *NO LCM* excel on F5, F11, and F19, respectively. This could be attributed to potential overfitting of POM to the relatively simple training set. The exclusion of mask operation (*NO MASK*) significantly diminishes POM's performance, highlighting the importance of the mask for global information sharing and population interaction, crucial for maintaining diversity. All modules contribute to POM's overall performance, with the negative impact on POM's performance ranked as follows: *NO MASK > NO LMM > UNTRAINED > NO LCM*.

**Fine-tuning Test**    We evaluate the fine-tuned POM's performance on $TF6$-$TF8$ as detailed in Appendix Table 3. We replace LCM with a standard transformer encoder to obtain more stable experimental results. For $\mathbf{x}_i$ and $\mathbf{v}_i$, we normalize their features and then concatenate them by dimension to obtain $\mathbf{xv}_i \in \mathbb{R}^{d \times 2}$. $\mathbf{xv}_i$ and the normalized [fitness, ranking] information of $\mathbf{x}_i$ in the parent population are concatenated to obtain $\mathbf{xvf}_i \in \mathbb{R}^{(d+1) \times 2}$. Based on this input, the transformer encoder will generate $\mathbf{cv}_i$ in Eq. (12). Different numbers of $\omega$ are generated as fine-tuning samples for $TF6$-$TF8$, and Algorithm 1 is used to fine-tune POM for each function. The base POM is

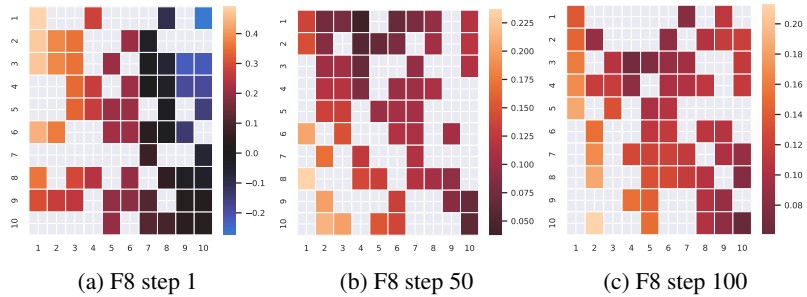

(a) F8 step 1          (b) F8 step 50          (c) F8 step 100

Figure 8: Displayed are visualized outcomes of LMM $S^t$ in BBOB with $d = 100$ using $n = 10$ for clarity. Blank squares in the matrix denote masked portions from Eq. (8). Steps 1, 50, and 100 correspond to the 1st, 50th, and 100th generations in population evolution. The horizontal and vertical axes denote individual rankings, with 1 as the best and 10 as the worst in the population. Each row illustrates the weight assigned to other individuals when executing mutation operations for the respective individual.

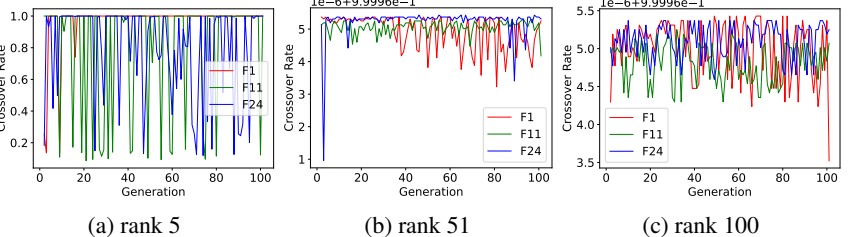

(a) rank 5          (b) rank 51          (c) rank 100

Figure 9: Visual analysis results of LCM on BBOB F1, F11, and F24 with $d = 100$, employing $n = 100$, are presented. "Rank" signifies an individual's position, with rank 5 representing the fifth-ranked individual in the population. Subgraphs depict the evolution of the probability that an individual will undergo crossover across three tasks as the population progresses. For example, (a) illustrates the crossover probability change for the top-ranked individual on F1, F11, and F24 with the number of generations.

initially trained on $TF1$-$TF5$. We calculate the relative performance improvement (RFI) achieved by the fine-tuned POM compared to the base POM, with results displayed in Figure 6. Experimental results indicate that fine-tuning POM leads to significant performance improvements even with a small sample size. The method for obtaining fine-tuning samples is not restricted; for black-box tasks, a surrogate model can be constructed to facilitate fine-tuning.

**Size of Training Dataset** Any complex problem can be simulated by a polynomial composed of simple basic function terms. To ensure that the optimization strategy learned by POM has robust generalization ability and performance, we should train POM on a set of basic functions.

First, we tested the impact of increasing the number of basic functions in the training set on model performance. Next, we examined the effect of introducing complex functions into the training set. Functions $TF1 - TF5$ are basic simple terms. For example, $TF1$ is an absolute value term, and $TF5$ is a square summation term. Functions $TF6 - TF8$ are composite terms composed of several basic functions. For instance, $TF6$ includes both a cumulative multiplier term and a cosine term. The test results are shown in Figure 5 (a) and (b) (see Appendix Table 8 for details), respectively.

Experimental results indicate that increasing the number of basic functions leads to an overall improvement in POM performance, whereas the introduction of composite terms results in a significant performance decline. This aligns with our hypothesis.

**Scale of POM** We explore the performance of POM at different scales, which is shown in Fig. 4 (b) (refer to Appendix Table 9 for additional details). We increase POM's parameter count by perturbing the hidden layers of each module $(d_m, d_c)$. Six models are constructed in ascending order of parameter count, labeled as *VS* (very small), *S* (small), *M* (medium), *L* (large), *VL* (very large), and *XL* (extra large) (details in the Appendix Table 2). *XL* achieves the best performance, while *VS* and *M* also perform well. *S* exhibits the worst performance, and *VL* performs worse than *L*. Two core factors contribute to this phenomenon: the number of parameters and training. We observe a complex relationship between the number of parameters and training difficulty. *VS*, with the fewest

parameters, is the easiest to train and performs well on BBOB. Conversely, *XL*, with a large number of parameters, exhibits the strongest capability to represent strategies, resulting in the best performance. The performance of *XL* aligns with our expectations. We obtain the following principles: 1) Larger models can have stronger capabilities but are more challenging to train; 2) Training difficulty and model scale do not exhibit a simple linear relationship, warranting further research; 3) Larger models require more functions for effective training.

**Time Budget**   We assess the training and test time efficiency of POM across various architectures on BBOB ($d = 10$) and BBOB ($d = 100$) respectively, as illustrated in Figure 7. POM demonstrates remarkable efficiency in tackling BBO problems, with negligible training costs relative to its exceptional generalization ability and high performance.

### 4.4   Visualization Analysis

**LMM Learning Analysis**   Figure 8 displays $S^t$ for an in-depth analysis of the LMM strategy (refer to Appendix Figure 15-20 for additional details). Key observations and conclusions include: 1) Generally, superior individuals receive higher weights during LMM, showcasing POM's ability to balance exploration and exploitation as the population converges. 2) Across diverse function problems, POM dynamically generates optimization strategies, highlighting its adaptability and contributing to robust generalization. 3) Disadvantaged individuals exhibit a more uniform weight distribution, potentially aiding in their escape from local optima and enhancing algorithm convergence.

**LCM Learning Analysis**   We visually examine the LCM strategy, presenting the results in Fig. 9 (refer to Appendix Figure 21-26 for additional details). LCM displays the capacity to adaptively generate diverse strategies for individuals across different ranks in the population, revealing distinct patterns among tasks and rankings. Notably, top-ranking individuals within the top 20, such as those ranked 1st, 5th, and 18th, exhibit a flexible crossover strategy. The dynamic adjustment of crossover probability with population evolution aids in preserving dominant genes and facilitating escape from local optima. Conversely, lower-ranking individuals show an increasing overall probability of crossover, promoting exploration of disadvantaged individuals and enhancing the algorithm's exploration capability. LCM proficiently generates adaptive crossover strategies across tasks, individuals, and convergence stages, significantly boosting both convergence and exploration capabilities.

## 5   Conclusions

We present POM, a novel Pretrained Optimization Model designed to address the inefficiencies of existing methods in zero-shot optimization. Evaluation on BBOB and robot control tasks demonstrates POM's superiority over other black-box optimizers, particularly in high-dimensional scenarios. Additionally, POM excels in solving few-shot optimization problems. Future research avenues include designing enhanced loss functions to optimize POM for both population convergence and diversity, thereby improving overall algorithm performance. In addition, the limitations of model scale and time performance deserve further study (see Appendix I for details).

## Acknowledgements

This work was supported in part by the National Natural Science Foundation of China under Grant 62206205 and 62471371, in part by the Young Talent Fund of Association for Science and Technology in Shaanxi, China under Grant 20230129, in part by the Guangdong High-level Innovation Research Institution Project under Grant 2021B0909050008, and in part by the Guangzhou Key Research and Development Program under Grant 202206030003.

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

# A Preliminaries

## A.1 Genetic Algorithms

The crossover, mutation, and selection operators form the basic framework of GAs. GA starts with a randomly generated initial population. Then, genetic operations such as crossover and mutation will be carried out. After the fitness evaluation of all individuals in the population, a selection operation is performed to identify fitter individuals to undergo reproduction to generate offspring. Such an evolutionary process will be repeated until specific predefined stopping criteria are satisfied.

**Crossover**   The crossover operator generates a new individual $\mathbf{x}_i^c \in \mathbb{R}^d$ by Eq. (17), and $cr$ is the probability of the crossover operator.

$$x_k^c = \begin{cases} x_k^i & rand(0,1) < cr \\ x_k^j & otherwise \end{cases} \tag{17}$$

where $k \in [1, \cdots, d]$, $i$ and $j \in [1, 2, \ldots, n]$ $(i \neq j)$. $d$ represents the dimension of the problem. $x_k^i$ and $x_k^j$ represent the $k$-th element of $\mathbf{x}^i$ and $\mathbf{x}^j$ respectively. This operator is commonly conducted on $n$ individuals; $n$ represents the population size. After an expression expansion, we reformulate Eq. (17) as $\sum_{i=1}^n \mathbf{x}_i \mathbf{W}_i^c$ [46]. $\mathbf{x}_i \in \mathbb{R}^d$ represents the $i$th individual in $\mathbf{X}$, where $\mathbf{X} = \{\mathbf{x}_1, \mathbf{x}_2, \cdots, \mathbf{x}_n\}$ is a population. $\mathbf{W}_i^c \in \mathbb{R}^{d \times d}$ is the diagonal matrix. If $\mathbf{W}_i^c$ is full of zeros, the $i$th individual has no contribution.

**Mutation**   The mutation operator brings about random changes in the population. Specifically, an individual $\mathbf{x}_i$ in the population goes through the mutation operator to form the new individual $\mathbf{x}_i^m$, formulated as follows:

$$x_k^m = \begin{cases} rand(l_k, u_k) & rand(0,1) < mr \\ x_k^c & otherwise \end{cases} \tag{18}$$

where $mr$ is the probability of mutation operator and $k \in [1, \cdots, d]$. $x_k^m$ and $x_k^c$ represent the $k$-th element of $\mathbf{x}^m$ and $\mathbf{x}^c$ respectively. Similarly, Equation (18) can be reformulated as $\mathbf{x}_i^c \mathbf{W}_i^m$, where $\mathbf{W}_i^m \in \mathbb{R}^{d \times d}$ is the diagonal matrix.

**Selection**   We introduce the binary tournament mating selection operator in Eq. (19). The selection operator survives individuals of higher quality for the next generation until the number of individuals is chosen. As shown in Eq. (19),

$$p_i = \begin{cases} 1 & f(\mathbf{x}_i) < f(\mathbf{x}_k) \\ 0 & f(\mathbf{x}_i) > f(\mathbf{x}_k) \end{cases}, \quad (\mathbf{x}_i, \mathbf{x}_k) \in \mathbf{X} \cup \mathbf{X}^m \tag{19}$$

where $p_i$ reflects the probability that $\mathbf{x}_i$ is selected for the next generation, and $\mathbf{X}^m = \{\mathbf{x}_1^m, \mathbf{x}_2^m, \cdots, \mathbf{x}_n^m\}$.

## A.2 Mutation Strategy in DE

The core components of the optimization model include modules that generate solutions and modules that select solutions. GA and DE basically include crossover modules, mutation modules and selection modules. The evolutionary strategy represented by CMA-ES needs to sample a population from a certain distribution (such as Gaussian distribution), and further select individuals to update this distribution. In this paper, we design parameterized trainable LMM and LCM as modules for generating solutions. The function of LMM is to generate a candidate population, and LCM further performs crossover between the candidate population and the original population to obtain the offspring population.

We list some classic DE mutation strategies.

- DE/rand/1

$$\mathbf{v}_i^t = \mathbf{x}_{r1}^t + F \cdot (\mathbf{x}_{r2}^t - \mathbf{x}_{r3}^t) \tag{20}$$

- DE/rand/2

$$\mathbf{v}_i^t = \mathbf{x}_{r1}^t + F \cdot (\mathbf{x}_{r2}^t - \mathbf{x}_{r3}^t + \mathbf{x}_{r4}^t - \mathbf{x}_{r5}^t) \tag{21}$$

- DE/best/1

$$\mathbf{v}_i^t = \mathbf{x}_{best}^t + F \cdot (\mathbf{x}_{r1}^t - \mathbf{x}_{r2}^t) \tag{22}$$

- DE/current-to-rand/1

$$\mathbf{v}_i^t = (1 - F)\mathbf{x}_i^t + F \cdot (\mathbf{x}_{r1}^t - \mathbf{x}_{r2}^t + \mathbf{x}_{r3}^t) \tag{23}$$

- DE/current-to-best/1

$$\mathbf{v}_i^t = (1 - F)\mathbf{x}_i^t + F \cdot \mathbf{x}_{best}^t + F \cdot (\mathbf{x}_{r1}^t - \mathbf{x}_{r2}^t) \tag{24}$$

- DE/current-to-pbest/1

$$\mathbf{v}_i^t = (1 - F)\mathbf{x}_i^t + F \cdot \mathbf{x}_{pbest}^t + F \cdot (\mathbf{x}_{r1}^t - \mathbf{x}_{r2}^t) \tag{25}$$

The integer index $r1$ (and similarly, $r2$ and $r3$) is randomly selected from the range $[0, N]$. $pbest$ is randomly selected from the indices of the best $p$ individuals. $x_{best}^t$ is the individual with the best fitness in the population at generation $t$.

The generalized form of the mutation strategy is

$$\mathbf{v}_i^t = \sum_j^N w_{i,j} \mathbf{x}_j \quad (\forall w_{i,j} \in \mathbb{R}) \tag{26}$$

For example, when $w_{i,q} = 1, w_{i,k} = -w_{i,j} \neq 0$, and $w_{i,l} = 0$ ($\forall l \notin \{q, j, k\}, q \neq k, k \neq j, q \neq j$), it becomes DE/rand1/1. If individuals of the population has been sorted from good to bad by fitness, when $w_{i,0} = 1, w_{i,k} = -w_{i,j} \neq 0$, and $w_{i,l} = 0$ ($\forall l \notin \{0, j, k\}, k \neq j$), it becomes DE/best/1.

## B  Landscape Features of TF1-TF8

The landscape features included in $TS$ are shown as follows:

- TF1: Unimodal
- TF2: Separable
- TF3: Unimodal, Separable
- TF4: Unimodal, Separable
- TF5: Multimodal, Non-separable, Having a very narrow valley from local optimum to global optimum, Ill-conditioned
- TF6: Multi-modal, Non-separable, Rotated
- TF7: Multimodal, Separable, Asymmetrical, Local optima's number is huge
- TF8: Multi-modal, Non-separable, Asymmetrical

## C  Test Set

### C.1  BBOB

BBOB [59, 55] is a widely researched and recognized collection of benchmark test problems to evaluate the performance of optimization algorithms. The dataset consists of a series of high-dimensional continuous optimization functions, including single-peak, multi-peak, rotated, and distorted functions, as well as some functions with specific properties such as Lipschitz continuity and second-order differentiability.

### C.2  Robot Control Tasks

We test the performance of POM on two complex robot control tasks.

### C.2.1 Bipedal Walker

The continuous control task Bipedal Walker [58], implemented within the Box2D physics engine, has been designed to test the ability of walking agents to navigate varying terrain by controlling their joints and maintaining balance. The challenge requires the agent to learn efficient walking strategies that enable it to traverse the intended path without falling or deviating from its trajectory. The robot's state comprises a range of variables, including the hull angle speed, angular velocity, horizontal speed, vertical speed, joint positions and angular speeds, legs contact with the ground, and lidar rangefinder measurements. The robot's actions involve determining motor speed values in the range of [-1, 1] for each of the four joints at the hips and knees. The performance of the agent is evaluated through a reward system, whereby it receives points for moving forward, with a maximum of 300+ points awarded upon successfully reaching the end of the designated course. However, the penalty of -100 points is imposed if the robot loses balance and falls. Furthermore, applying motor torque incurs a small cost in terms of points. The score accrued by the agent serves as a measure of its optimal performance. The Bipedal Walker task represents a challenging and dynamic environment that effectively evaluates the walking and balance control abilities of agents. As such, it provides a valuable benchmark for testing and comparing different reinforcement learning algorithms for robotic locomotion.

### C.2.2 Enduro

Enduro [58] is one of the classic reinforcement learning environments provided by OpenAI Gym. It is a driving racing game based on the Atari 2600 game. In this environment, your goal is to drive as far as possible by controlling the car. The Enduro game is set on an endless highway where you need to avoid other vehicles and overtake as many other vehicles as possible within a limited time. You can avoid collisions with other vehicles by moving your car left and right, and be careful to control your speed to avoid accidents. The game rewards you based on how far you drive, so your goal is to learn a good driving strategy to maximize the distance traveled.

In these two test tasks, the agent interacts with the environment for $k$ time steps, and the reward at the $i$-th step is $r_i$. We evaluate strategy performance as follows:

$$R = \sum_{i=0}^{k} r_i \tag{27}$$

In these two tasks we conduct 10 sets of experiments, each set of experiments consists of 5 independent runs. We finally take the best results of each set of experiments to calculate the mean and standard deviation.

## D   Baselines

Our core is the population-based pre-training BBO algorithm, so we do not compare with non-population methods such as Bayesian optimization methods. Moreover, Bayesian optimization methods are difficult to deal with continuous optimization problems of more than 100 dimensions. We do not use LLM-based approaches [28–32, 34–36] as baselines because they can only be used for a specific type of task.

**Heuristic Population-based BBO Algorithm**. DE(DE/rand/1/bin) [54], ES(($\mu,\lambda$)-ES), L-SHADE [22], and CMA-ES [12], where DE [54] and ES are implemented based on Geatpy [60], CMA-ES and IPOP-CMA-ES are implemented by cmaes[2], and L-SHADE is implemented by pyade[3]. The reasons for choosing these baselines are the following:

- DE(DE/rand/1/bin): A classic numerical optimization algorithm.
- ES(($\mu,\lambda$)-ES): A classic variant of the evolution strategy.
- CMA-ES: CMA-ES is often considered the state-of-the-art method for continuous domain optimization under challenging settings (e.g., ill-conditioned, non-convex, non-continuous, multimodal).
- L-SHADE: The state-of-the-art variant of DE.

**Pretrained BBO Algorithm**. We chose three state-of-the-art meta-learn BBO algorithms for comparison with POM.

- LES [25]: A recently proposed learnable ES. It uses a data-driven approach to discover new ES with strong generalization performance and search efficiency.
- LGA [10]: A recently proposed learnable GA that discovers new GA in a data-driven manner. The learned algorithm can be applied to unseen optimization problems, search dimensions, and evaluation budgets.
- We train POM on $TS$. During training, the maximum number of evolution generations is 100, $n = 100$ and the problem dimension is set to 10.

---

[2]https: //github.com/CyberAgentAILab
[3]https://github.com/xKuZz/pyade

# E  Parameters and Training Dataset

The primary control parameters of CMA-ES and L-SHADE are automatically adjusted. For LGA and LES, we utilized the optimal parameters provided by the authors without modifications. Other hyperparameters were tuned using grid search to identify the optimal combinations, and multiple experiments were conducted accordingly. Detailed parameter settings are presented in Table 1. Each experiment reports the mean and standard deviation of the results from various sets of experiments, with a consistent population size of 100 across all trials. All experiments are performed on a device with GeForce RTX 3090 24G GPU, Intel Xeon Gold 6126 CPU and 64G RAM.

Table 1: Detailed parameter settings for all baselines.

| Algorithm | item | setting |
|---|---|---|
| POM | $d_m = 1000$ 
 $d_c = 4$ | Standard Settings for POM (M). |
| CMA-ES | Initial $\sigma = \frac{\text{upper\_bounds}+\text{lower\_bounds}}{2} * \frac{2}{5}$ | $2/5$ is a hyperparameter, and we determine this hyperparameter between $[0.1, 1]$ using a grid search, with a step of 0.1. |
| | Initial $\mu$ | $\mu = \textbf{lower\_bounds} + (randn(d) * (\textbf{upper\_bounds} - \textbf{lower\_bounds}))$, where $randn(d)$ stands for sampling a $d$-dimensional vector from a standard normal distribution. |
| LSHADE | $memory\_size = 6$ | We use a grid search to determine this parameter, the search interval is $[1, 10]$, and the search step is 1. |
| ES | $selFuc = urs$ | We use a grid search to determine this parameter, the search interval is $[dup, ecs, etour, otos, rcs, rps, rws, sus, tour, urs]$ [60]. |
| | $Nsel = 0.5$ | we determine this hyperparameter between $[0.1, 0.8]$ using a grid search, with a step of 0.1. |
| DE | $F = 0.5$ | we determine this hyperparameter between $[0.1, 0.9]$ using a grid search, with a step of 0.1. [60]. |
| | $XOVR = 0.5$ | we determine this hyperparameter between $[0.1, 0.9]$ using a grid search, with a step of 0.1. |
| LGA | All parameters | We use the pre-trained optimal parameters provided by the authors. |
| LES | All parameters | We use the pre-trained optimal parameters provided by the authors. |

Table 2: POM parameters of different architectures and architecture settings.

| STRUCTURE | number of parameters | $d_m$ | $d_c$ |
|---|---|---|---|
| VS | 40929 | 200 | 4 |
| S | 101529 | 500 | 4 |
| M | 202529 | 1000 | 4 |
| L | 404641 | 2000 | 20 |
| VL | 110851 | 5000 | 50 |
| XL | 2021201 | 10000 | 100 |

Table 3: Additional Training Functions. $z_i = x_i - \omega_i$.

| ID | Functions | Range |
|---|---|---|
| TF1 | $\sum_i \lvert x_i - \omega_i\rvert$ | $x \in [-10, 10], \omega \in [-10, 10]$ |
| TF2 | $\sum_i \lvert (x_i - \omega_i) + (x_{i+1} - \omega_{i+1})\rvert + \sum_i \lvert x_i - \omega_i\rvert$ | $x \in [-10, 10], \omega \in [-10, 10]$ |
| TF3 | $\sum_i z_i^2$ | $x \in [-100, 100], \omega \in [-50, 50]$ |
| TF4 | $\max\{\lvert z_i\rvert, 1 \le i \le d\}$ | $x \in [-100, 100], \omega \in [-50, 50]$ |
| TF5(Rosenbrock) | $\sum_{i=1}^{d-1}(100(z_i^2 - z_{i+1})^2 + (z_i - 1)^2)$ | $x \in [-100, 100], \omega \in [-50, 50]$ |
| TF6(Griewank) | $\sum_{i=1}^{d} \frac{z_i^2}{4000} - \prod_{i=1}^{d}\cos(\frac{z_i}{\sqrt{i}}) + 1$ | $x \in [-600, 600], \omega \in [-300, 300]$ |
| TF7(Rastrigin) | $\sum_{i=1}^{d}(z_i^2 - 10\cos(2\pi z_i) + 10)$ | $x \in [-5, 5], \omega \in [-2.5, 2.5]$ |
| TF8(Ackley) | $-20\exp(-0.2\sqrt{\frac{1}{d}\sum_{i=1}^{d} z_i^2}) - \exp(\frac{1}{d}\sum_{i=1}^{d}\cos(2\pi z_i)) + 20 + \exp(1)$ | $x \in [-32, 32], \omega \in [-16, 16]$ |

# F  Additional Experimental results on BBOB

## F.1  BBOB Test

Table 4: BBOB RESULT. POM is trained on TF1-TF5 with $d$ =10. The best results are indicated in bold, and the suboptimal results are underlined.

| $d$ | F | POM | ES | DE | CMA-ES | LSHADE | LES | LGA |
|---|---|---|---|---|---|---|---|---|
| 30 | F1 | **3.72E-11(3.72E-11)** | 2.30E+02(1.36E+01) | 9.46E+01(1.17E+01) | 7.79E-04(8.97E-04) | 1.28E-03(7.36E-04) | 4.93E+00(4.03E+00) | 1.13E+01(5.81E+00) |
| | F2 | **4.69E-12(4.69E-12)** | 2.18E+00(5.24E-01) | 1.10E-01(4.35E-03) | 8.45E-02(1.64E-02) | 1.47E-05(2.12E-06) | 1.45E-02(5.21E-03) | 1.75E-01(4.80E-02) |
| | F3 | **6.57E+01(6.57E+01)** | 1.41E+03(1.26E+02) | 1.02E+03(5.47E+01) | 2.47E+03(2.39E+03) | 7.12E+01(9.31E+00) | 8.10E+02(1.04E+02) | 2.82E+02(1.66E+01) |
| | F4 | **6.95E+01(6.95E+01)** | 3.35E+03(6.76E+02) | 1.99E+03(5.61E+02) | 2.21E+02(1.02E+00) | 1.04E+02(4.24E+00) | 6.11E+01(1.22E+02) | 3.76E+02(3.14E+01) |
| | F5 | 3.61E+01(3.61E+01) | 5.52E+01(1.45E+01) | 1.32E+00(2.70E-01) | **0.00E+00(0.00E+00)** | 0.00E+00(0.00E+00) | 1.99E+02(4.46E+01) | 0.00E+00(0.00E+00) |
| | F6 | **1.69E-09(1.69E-09)** | 3.97E+02(9.66E+00) | 5.29E+02(1.74E+02) | 8.99E-02(6.01E-03) | 1.54E-01(8.94E-02) | 1.11E+01(7.44E+00) | 2.25E+01(5.33E+00) |
| | F7 | **3.78E-13(3.78E-13)** | 1.61E+03(5.19E+01) | 7.62E+03(9.02E+02) | 3.44E+00(7.67E-01) | 1.25E+01(6.46E+00) | 1.20E+02(3.92E+01) | 6.97E+01(2.00E+01) |
| | F8 | **6.23E-06(6.23E-06)** | 4.21E+05(6.85E+04) | 3.26E+05(4.66E+04) | 3.15E+02(3.90E+02) | 3.08E+01(3.53E+00) | 3.01E+03(2.28E+03) | 1.63E+03(2.94E+02) |
| | F9 | 1.60E+02(1.60E+02) | 4.44E+05(8.88E+04) | 7.06E+05(9.64E+04) | **4.17E+01(1.20E+01)** | 5.85E+01(5.42E+01) | 2.37E+03(6.93E+02) | 1.38E+03(4.40E+02) |
| | F10 | **2.24E+03(2.24E+03)** | 3.56E+06(1.48E+06) | 2.33E+07(6.30E+06) | 3.39E+05(1.18E+05) | 1.16E+04(5.72E+03) | 7.58E+04(2.93E+04) | 2.67E+05(5.13E+04) |
| | F11 | **7.38E+00(7.38E+00)** | 1.59E+03(5.31E+02) | 5.73E+03(8.62E+02) | 5.55E+03(1.21E+03) | 1.53E+02(1.13E+02) | 2.36E+02(2.59E+01) | 3.95E+02(1.40E+02) |
| | F12 | **5.13E-04(5.13E-04)** | 4.18E+09(4.62E+08) | 1.37E+10(8.87E+08) | 2.91E+11(2.89E+10) | 4.10E+05(4.53E+05) | 1.04E+08(6.51E+07) | 9.59E+07(2.87E+07) |
| | F13 | **6.76E-05(6.76E-05)** | 1.57E+03(6.29E+01) | 1.07E+03(8.97E+01) | 9.66E+00(1.62E+00) | 2.44E+00(1.41E+00) | 8.61E+01(3.33E+01) | 2.40E+02(3.31E+01) |
| | F14 | **2.29E-04(2.29E-04)** | 9.04E+01(1.08E+01) | 5.84E+02(6.23E+01) | 1.92E+00(1.34E+00) | 4.38E-02(2.39E-02) | 6.01E+00(1.54E+00) | 4.02E+00(7.88E-01) |
| | F15 | **7.84E+01(7.84E+01)** | 1.62E+03(1.29E+02) | 4.31E+03(6.26E+02) | 4.27E+04(3.66E+04) | 1.16E+02(1.08E+01) | 8.73E+02(1.65E+02) | 2.84E+02(1.90E+01) |
| | F16 | 2.55E+01(2.55E+01) | 4.62E+01(4.62E+00) | 5.44E+01(5.74E+00) | 3.18E+01(3.66E+00) | 1.64E+01(5.59E+00) | **7.17E+00(9.49E-01)** | 3.24E+01(8.73E-01) |
| | F17 | **2.79E-05(2.79E-05)** | 2.47E+01(7.36E+00) | 2.43E+01(4.93E+00) | 3.78E-01(8.36E-02) | 4.67E-01(1.09E+00) | 9.74E+00(3.39E+00) | 2.21E+00(2.95E-01) |
| | F18 | **1.30E-01(1.30E-01)** | 9.84E+01(1.68E+01) | 1.19E+02(4.66E+01) | 2.26E+00(5.51E-01) | 9.34E-01(3.55E-01) | 3.43E+01(1.02E+01) | 1.21E+01(2.63E+00) |
| | F19 | **4.82E+00(4.82E+00)** | 5.43E+01(4.16E+00) | 5.00E+01(1.17E+01) | 5.94E+00(4.07E-01) | 5.44E+00(4.67E-01) | 1.61E+01(2.11E+00) | 7.06E+00(2.09E-01) |
| | F20 | -1.32E+01(-1.32E+01) | 1.25E+05(3.14E+04) | 1.08E+05(2.55E+04) | 3.27E+00(1.03E-01) | 3.13E+00(9.10E-02) | **-2.72E+01(1.03E+01)** | 9.09E+01(8.60E+01) |
| | F21 | **3.36E-01(3.36E-01)** | 8.80E+01(6.16E-01) | 8.56E+01(7.64E-01) | **2.89E+00(5.34E-02)** | 1.44E+01(1.26E+01) | 1.99E+01(8.05E+00) | 9.98E+00(2.36E+00) |
| | F22 | 1.57E+01(1.57E+01) | 8.92E+01(1.82E+00) | 8.57E+01(6.39E-01) | 1.96E+00(5.02E-03) | **1.14E+00(7.17E-01)** | 1.68E+01(3.62E+00) | 9.91E+00(4.49E+00) |
| | F23 | 3.68E+00(3.68E+00) | 1.38E+01(8.94E-01) | 1.16E+01(1.80E+00) | 3.85E+00(3.62E-01) | 3.17E+00(8.65E-01) | **3.01E+00(3.26E-01)** | 4.38E+00(1.13E-01) |
| | F24 | 2.81E+02(2.81E+02) | 4.10E+04(6.39E+04) | 5.32E+04(4.52E+04) | 2.23E+02(5.47E+00) | **1.72E+02(5.53E+00)** | 7.08E+02(6.37E+01) | 3.69E+02(3.53E+01) |
| 100 | F1 | **5.92E-12(5.92E-12)** | 1.60E+03(3.45E+01) | 4.62E+03(5.31E+02) | 4.34E+01(4.29E+00) | 1.64E+01(8.78E-01) | 2.20E+02(4.07E+01) | 1.13E+02(1.69E+01) |
| | F2 | **4.70E-12(4.70E-12)** | 4.08E+01(6.19E+00) | 2.24E+01(3.00E+00) | 4.17E+01(9.20E+00) | 7.58E-02(4.38E-02) | 4.56E+00(1.06E+00) | 3.28E+00(4.70E-01) |
| | F3 | **1.07E-09(1.07E-09)** | 1.06E+04(7.18E+02) | 4.77E+04(2.87E+03) | 3.24E+04(8.39E+03) | 8.71E+02(9.08E+01) | 2.72E+03(1.55E+02) | 1.82E+03(4.78E+01) |
| | F4 | **1.39E-07(1.39E-07)** | 6.28E+04(7.31E+03) | 2.96E+05(2.45E+04) | 3.77E+03(3.11E+02) | 1.29E+03(1.69E+02) | 5.15E+03(1.02E+03) | 2.49E+03(1.23E+02) |
| | F5 | 3.04E+02(3.04E+02) | 2.03E+01(1.42E+01) | 4.64E+00(2.13E+00) | 1.63E+02(2.83E+02) | **3.98E+00(4.69E+00)** | 1.30E+03(8.74E+01) | 4.05E+00(3.67E+00) |
| | F6 | **9.32E-10(9.32E-10)** | 2.46E+03(2.04E+02) | 9.15E+03(2.22E+02) | 2.65E+02(1.05E+02) | 4.00E+01(5.60E+00) | 4.37E+02(4.07E+01) | 2.09E+02(9.05E+00) |
| | F7 | **2.42E-13(2.42E-13)** | 1.11E+04(2.21E+03) | 6.11E+04(4.18E+03) | 2.79E+03(3.55E+02) | 1.96E+02(7.71E+01) | 1.43E+03(3.21E+02) | 9.43E+02(2.72E+02) |
| | F8 | **3.01E-08(3.01E-08)** | 2.09E+07(2.75E+05) | 1.60E+08(2.16E+07) | 6.06E+04(2.47E+04) | 1.35E+04(7.05E+03) | 2.40E+05(1.18E+04) | 9.43E+04(5.31E+04) |
| | F9 | **6.41E+02(6.41E+02)** | 1.97E+07(1.59E+06) | 2.18E+08(2.65E+07) | 1.72E+03(5.89E+02) | 4.06E+03(9.38E+02) | 3.71E+05(1.41E+04) | 1.07E+05(2.57E+04) |
| | F10 | **2.34E+01(2.34E+01)** | 5.73E+07(1.15E+07) | 3.29E+08(1.13E+07) | 7.27E+07(4.91E+07) | 4.19E+05(3.99E+04) | 2.82E+06(1.01E+06) | 3.83E+06(5.67E+05) |
| | F11 | **1.71E+01(1.71E+01)** | 4.63E+03(5.42E+02) | 2.41E+04(2.95E+02) | 3.25E+04(5.30E+03) | 4.59E+02(8.92E+01) | 7.82E+02(3.81E+01) | 1.27E+03(1.67E+02) |
| | F12 | **1.43E-04(1.43E-04)** | 4.15E+10(1.75E+09) | 4.86E+11(8.89E+10) | 1.12E+12(5.61E+11) | 9.01E+08(4.73E+08) | 3.83E+09(2.93E+08) | 2.12E+09(1.07E+09) |
| | F13 | **7.23E-05(7.23E-05)** | 4.18E+03(8.22E+01) | 6.65E+03(4.61E+02) | 6.35E+02(1.15E+02) | 3.89E+02(6.17E+01) | 1.53E+03(1.03E+02) | 9.26E+02(6.39E+01) |
| | F14 | **9.07E-05(9.07E-05)** | 4.51E+02(6.12E+01) | 3.85E+03(5.14E+02) | 4.15E+02(6.83E+01) | 7.45E+00(3.08E+00) | 3.57E+01(5.43E+00) | 4.11E+01(4.25E+00) |
| | F15 | **4.88E+02(4.88E+02)** | 9.88E+03(7.02E+02) | 6.86E+04(8.80E+03) | 3.31E+04(1.93E+04) | 1.05E+03(9.66E+01) | 3.61E+03(2.38E+02) | 1.65E+03(1.10E+01) |
| | F16 | 4.72E+01(4.72E+01) | 8.41E+01(3.87E+00) | 1.90E+02(1.40E+01) | 5.36E+01(3.48E+00) | 3.44E+01(3.21E+00) | **1.29E+01(5.22E-01)** | 5.58E+01(1.36E+00) |
| | F17 | **5.50E-07(5.50E-07)** | 1.26E+03(3.78E+02) | 1.77E+04(8.83E+02) | 5.71E+00(1.24E+00) | 2.61E+00(9.12E-02) | 2.10E+01(2.18E+00) | 1.20E+01(1.09E+00) |
| | F18 | **5.94E-06(5.94E-06)** | 1.73E+03(1.13E+02) | 2.66E+04(3.33E+03) | 2.65E+01(4.37E+00) | 1.06E+01(1.25E+00) | 5.66E+01(9.50E+00) | 4.16E+01(4.47E+00) |
| | F19 | **6.74E+00(6.74E+00)** | 5.37E+02(5.46E+01) | 5.32E+02(3.25E+01) | 1.08E+01(1.71E+00) | 8.95E+00(2.98E-01) | 2.75E+02(1.74E+00) | 1.30E+01(1.12E+00) |
| | F20 | **-5.08E+00(-5.08E+00)** | 1.56E+06(8.58E+04) | 5.16E+06(5.28E+05) | 3.98E+04(1.36E+04) | 1.70E+03(3.56E+02) | 4.66E+04(1.65E+04) | 2.56E+04(6.50E+03) |
| | F21 | 4.03E+01(4.03E+01) | 2.10E+02(4.08E+01) | 1.22E+03(2.28E+02) | 6.56E+01(1.25E+01) | **1.37E+01(3.04E+00)** | 7.62E+01(6.40E-01) | 7.35E+01(7.57E-01) |
| | F22 | 5.95E+01(5.95E+01) | 2.33E+02(1.93E+01) | 1.46E+03(4.77E+02) | 7.02E+01(2.84E+00) | **3.12E+01(1.82E+01)** | 7.62E+01(4.80E+00) | 8.01E+01(6.36E+00) |
| | F23 | **4.83E+00(4.83E+00)** | 1.39E+02(8.62E+00) | 1.04E+03(1.59E+02) | 5.78E+00(5.92E-01) | 5.02E+00(3.81E-01) | 5.21E+00(1.55E-01) | 7.15E+00(1.37E-01) |
| | F24 | **1.24E+03(1.24E+03)** | 1.32E+07(2.48E+06) | 1.30E+08(1.61E+07) | 1.61E+03(6.24E+01) | 1.24E+03(2.28E+01) | 3.52E+03(3.10E+02) | 3.37E+03(2.41E+02) |
| +/=/- | - | -/-/- | 47/0/1 | 46/0/2 | 41/1/6 | 35/2/11 | 43/0/5 | 44/0/4 |

## F.2 BBOB Test With Optimal Solution Disturbed

We further test the performance of the algorithm on the BBOB. Here, the optimal solution of each function is randomly disturbed, that is, $\mathbf{x}^* = \mathbf{x}_{opt} + \mathbf{z}$, where $\mathbf{x}^*$ represents the optimal solution after disturbing, $\mathbf{x}_{opt}$ represents the original optimal solution, $\mathbf{x}_{opt}$ is a vector obtained by random sampling and $\mathbf{z} \in [-1, 1]^d$. The results are displayed in Table 5 and Figure 10. We found that the performance of POM can still dominate other algorithms when the function optimal solution is disturbed.

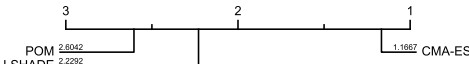

Figure 10: Critical difference diagram of 3 algorithms on 24 BBOB problems with $d = 100$. The locations of the optimal solutions are in the range of $[-1, 1]$.

Table 5: Additional Experimental results on BBOB ($d = 100$). The best results are indicated in bold, and the suboptimal results are underlined.

| F | POM | ES | DE | CMA-ES | LSHADE | LES | LGA |
|---|---|---|---|---|---|---|---|
| F1 | **1.85E-11(1.85E-11)** | 2.49E+02(1.29E+01) | 6.27E+02(7.89E+01) | 5.76E+00(1.69E+00) | 2.94E+00(8.22E-01) | 2.20E+02(9.01E+00) | 8.95E+01(1.19E+01) |
| F2 | **3.91E-12(3.91E-12)** | 5.95E+00(1.05E+00) | 4.31E+00(1.50E-01) | 6.51E+00(3.23E+00) | 1.43E-02(6.93E-03) | 3.38E+00(1.48E+00) | 3.39E+00(6.19E-01) |
| F3 | **9.64E+01(9.64E+01)** | 2.14E+03(6.76E+01) | 4.45E+03(3.99E+02) | 1.26E+03(5.32E+01) | 5.25E+02(3.82E+01) | 2.68E+03(1.15E+02) | 1.71E+03(1.32E+02) |
| F4 | **6.10E-08(6.10E-08)** | 3.82E+03(1.69E+02) | 1.40E+04(3.22E+03) | 1.68E+03(1.23E+02) | 7.09E+02(2.40E+01) | 5.24E+03(1.18E+03) | 2.76E+03(3.12E+02) |
| F5 | 3.50E+02(3.50E+02) | 7.94E+01(1.34E+01) | 2.87E+01(7.49E+00) | 2.22E+02(3.84E+02) | **3.62E+00(4.75E+00)** | 1.38E+03(1.12E+01) | 5.48E+00(7.74E+00) |
| F6 | **2.05E-10(2.05E-10)** | 4.55E+02(1.58E+01) | 1.53E+03(1.66E+02) | 5.10E+01(2.66E+01) | 7.96E+00(1.51E+00) | 3.99E+02(1.13E+01) | 2.70E+02(8.24E+01) |
| F7 | **3.22E-13(3.22E-13)** | 1.91E+03(1.57E+02) | 8.81E+03(8.16E+02) | 7.20E+02(2.23E+02) | 4.36E+01(9.36E+00) | 1.33E+03(5.42E+02) | 8.93E+02(1.21E+02) |
| F8 | **1.89E-08(1.89E-08)** | 5.54E+05(2.29E+04) | 4.55E+06(5.34E+05) | 3.02E+03(6.65E+02) | 8.35E+02(6.27E+01) | 3.96E+05(1.21E+05) | 2.37E+05(5.39E+04) |
| F9 | **6.39E+02(6.39E+02)** | 5.11E+05(3.79E+04) | 4.73E+06(6.75E+05) | 4.07E+03(1.11E+03) | 7.94E+02(1.25E+02) | 3.83E+05(5.75E+04) | 9.15E+04(1.63E+04) |
| F10 | **1.58E+03(1.58E+03)** | 9.00E+06(9.13E+05) | 4.71E+07(2.44E+06) | 1.49E+07(7.37E+06) | 6.95E+04(2.16E+04) | 2.14E+06(6.53E+05) | 3.03E+06(6.19E+05) |
| F11 | **2.14E+01(2.14E+01)** | 7.92E+02(1.49E+02) | 3.79E+03(2.36E+02) | 5.21E+03(2.43E+02) | 7.69E+01(1.13E+01) | 7.65E+02(5.37E+01) | 1.36E+03(2.93E+02) |
| F12 | **1.06E-04(1.06E-04)** | 3.97E+09(6.42E+07) | 2.98E+10(4.86E+08) | 1.51E+09(3.66E+08) | 6.04E+07(1.93E+07) | 3.70E+09(6.08E+08) | 2.25E+09(3.85E+08) |
| F13 | **5.51E-05(5.51E-05)** | 1.61E+03(2.89E+01) | 2.64E+03(1.44E+02) | 2.52E+01(1.45E+01) | 1.49E+02(1.06E+01) | 1.49E+03(3.60E+01) | 1.02E+03(2.22E+02) |
| F14 | **5.05E-05(5.05E-05)** | 5.67E+01(1.52E+00) | 4.11E+02(2.83E+01) | 5.52E+01(1.26E+01) | 1.05E+00(4.57E-01) | 3.85E+01(3.60E+00) | 4.21E+01(3.73E+00) |
| F15 | **5.11E+02(5.11E+02)** | 2.16E+03(2.43E+01) | 6.55E+03(6.17E+02) | 1.27E+03(3.28E+01) | 6.41E+02(6.21E+01) | 3.23E+03(3.14E+02) | 1.70E+03(1.76E+02) |
| F16 | 4.84E+01(4.84E+01) | 5.14E+01(1.67E+00) | 7.31E+01(5.97E+00) | 5.25E+01(1.70E+00) | 3.85E+01(4.42E+00) | **1.48E+01(3.28E+00)** | 5.31E+01(2.25E+00) |
| F17 | **5.90E-07(5.90E-07)** | 9.25E+00(4.82E-01) | 1.98E+02(6.26E+01) | 2.43E+00(3.89E-01) | 1.14E+00(1.70E-01) | 1.24E+01(7.44E-01) | 1.02E+01(6.05E-01) |
| F18 | **7.01E-06(7.01E-06)** | 3.54E+01(3.01E-01) | 3.04E+02(1.64E+02) | 9.59E+00(1.51E+00) | 3.74E+00(1.00E+00) | 6.79E+01(2.00E+01) | 3.39E+01(2.04E+00) |
| F19 | **7.07E+00(7.07E+00)** | 2.15E+01(8.02E-01) | 1.54E+02(3.80E+01) | 8.39E+00(2.80E-01) | 7.52E+00(1.58E-01) | 3.46E+01(1.48E+00) | 1.33E+01(1.28E+00) |
| F20 | **-3.43E+00(-3.43E+00)** | 1.03E+05(9.88E+03) | 6.61E+05(3.47E+04) | 1.87E+02(1.78E+02) | 3.66E+00(3.87E-02) | 6.16E+04(1.48E+04) | 6.31E+04(7.62E+03) |
| F21 | 6.40E+01(6.40E+01) | 8.29E+01(1.20E+00) | 1.03E+02(6.73E+00) | 2.46E+01(2.67E+00) | **1.21E+01(2.52E+00)** | 7.90E+01(1.46E+00) | 7.15E+01(7.37E+00) |
| F22 | 5.91E+01(5.91E+01) | 8.04E+01(2.71E+00) | 9.45E+01(3.28E+00) | 1.95E+01(3.81E+00) | **1.32E+01(2.31E+00)** | 7.81E+01(5.26E-01) | 7.79E+01(6.09E+00) |
| F23 | 5.05E+00(5.05E+00) | 6.57E+00(3.51E-01) | 1.61E+01(5.17E+00) | 5.51E+00(2.24E-01) | **4.94E+00(1.06E-01)** | 5.09E+00(3.62E-01) | 6.36E+00(5.37E-01) |
| F24 | 1.31E+03(1.31E+03) | 2.57E+03(2.56E+02) | 1.44E+06(1.77E+05) | 1.17E+03(7.12E+01) | 9.63E+02(1.01E+02) | 3.57E+03(1.69E+02) | 3.53E+03(1.90E+02) |
| win/tie/loss | -/-/- | 23/0/1 | 23/0/1 | 20/0/4 | 17/3/4 | 21/2/1 | 21/2/1 |

### F.3 Higher-Dimensional BBOB Test

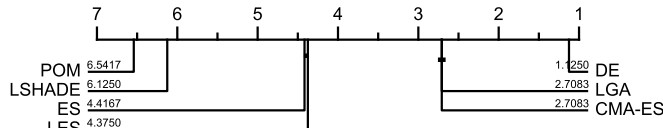

Figure 11: Critical difference diagram of 7 algorithms on 24 BBOB problems with $d = 500$.

Table 6: Additional Experimental results on BBOB ($d = 500$). The best results are indicated in bold, and the suboptimal results are underlined.

| F | POM | ES | DE | CMA-ES | LSHADE | LES | LGA |
|---|---|---|---|---|---|---|---|
| F1 | **1.98E-12(1.98E-12)** | 2.31E+03(6.40E+01) | 6.15E+03(6.55E+02) | 2.48E+03(8.03E+01) | 1.28E+02(1.87E+01) | 2.74E+03(5.65E+01) | 4.16E+03(5.26E+01) |
| F2 | **2.53E-12(2.53E-12)** | 9.56E+01(7.06E+00) | 2.28E+02(2.16E+01) | 2.11E+02(1.23E+01) | 1.95E+00(2.56E-01) | 7.48E+01(3.45E+00) | 9.80E+01(1.12E+01) |
| F3 | **2.33E-11(2.33E-11)** | 1.67E+04(2.48E+02) | 5.27E+04(2.96E+03) | 3.02E+04(8.66E+02) | 3.53E+03(4.12E+02) | 1.91E+04(3.75E+02) | 2.92E+04(1.15E+03) |
| F4 | **2.47E-09(2.47E-09)** | 5.06E+04(4.80E+03) | 2.79E+05(3.05E+04) | 8.90E+04(1.14E+04) | 6.90E+03(1.38E+03) | 1.31E+05(1.28E+04) | 1.66E+05(1.40E+04) |
| F5 | 4.97E+03(4.97E+03) | 3.07E+03(9.94E+01) | 3.92E+03(1.36E+02) | **1.26E+03(1.95E+02)** | 2.68E+03(2.48E+02) | 8.44E+03(3.08E+02) | 2.86E+03(8.51E+01) |
| F6 | **8.96E-11(8.96E-11)** | 3.91E+03(1.55E+02) | 9.14E+03(9.31E+01) | 4.51E+03(1.07E+02) | 2.13E+02(2.23E+01) | 3.93E+03(1.58E+02) | 5.63E+03(1.75E+02) |
| F7 | **1.60E-13(1.60E-13)** | 1.70E+04(1.91E+02) | 5.54E+04(2.24E+03) | 2.70E+04(1.34E+03) | 8.52E+02(1.20E+02) | 2.14E+04(1.58E+03) | 2.96E+04(1.03E+03) |
| F8 | **3.75E-08(3.75E-08)** | 2.27E+08(5.75E+06) | 1.57E+09(3.05E+07) | 2.80E+08(1.56E+07) | 8.02E+05(1.97E+05) | 1.69E+08(1.41E+07) | 4.64E+08(1.05E+07) |
| F9 | **3.24E+03(3.24E+03)** | 2.07E+08(1.80E+07) | 1.46E+09(2.76E+08) | 2.85E+08(1.61E+07) | 5.65E+05(1.21E+05) | 2.46E+08(1.50E+07) | 6.10E+08(2.26E+07) |
| F10 | **2.71E-05(2.71E-05)** | 1.15E+08(6.37E+06) | 4.57E+08(3.63E+07) | 2.36E+08(3.43E+07) | 2.48E+06(6.76E+05) | 9.21E+07(1.18E+07) | 9.36E+07(2.24E+07) |
| F11 | **1.17E+02(1.17E+02)** | 4.09E+03(2.27E+02) | 1.79E+04(2.97E+03) | 2.46E+04(1.39E+03) | 1.42E+03(5.16E+02) | 4.15E+03(1.31E+02) | 5.36E+03(4.49E+02) |
| F12 | **2.12E-05(2.12E-05)** | 4.52E+10(6.07E+08) | 1.09E+12(2.33E+11) | 2.28E+11(4.24E+10) | 1.48E+09(1.87E+08) | 4.90E+10(6.16E+08) | 1.69E+11(5.93E+09) |
| F13 | **1.29E-04(1.29E-04)** | 4.86E+03(5.22E+01) | 7.87E+03(3.62E+02) | 4.89E+03(3.96E+01) | 1.12E+03(1.31E+01) | 5.28E+03(1.04E+02) | 6.41E+03(1.02E+02) |
| F14 | **1.59E-06(1.59E-06)** | 2.76E+02(2.14E+01) | 1.79E+03(8.21E+01) | 6.18E+03(1.89E+01) | 1.15E+01(8.01E-01) | 2.48E+02(4.01E+01) | 5.33E+02(3.10E+01) |
| F15 | **1.15E+02(1.15E+02)** | 1.63E+04(4.57E+02) | 5.61E+04(5.73E+02) | 2.50E+04(2.12E+03) | 4.66E+03(1.29E+02) | 2.12E+04(2.15E+03) | 2.84E+04(1.07E+03) |
| F16 | 6.52E+01(6.52E+01) | 6.56E+01(2.66E+00) | 8.84E+01(4.04E-01) | 7.60E+01(2.29E+00) | 5.63E+01(1.15E+00) | **2.92E+01(1.06E+00)** | 7.04E+01(1.29E+00) |
| F17 | **2.56E-07(2.56E-07)** | 8.27E+01(6.17E+00) | 3.86E+03(4.66E+02) | 4.11E+02(8.71E+01) | 1.80E+00(1.06E-01) | 1.96E+01(1.30E+00) | 1.73E+02(2.50E+01) |
| F18 | **2.75E-07(2.75E-07)** | 1.35E+02(2.64E+01) | 3.10E+03(8.03E+02) | 3.77E+02(4.29E+01) | 7.58E+00(3.17E-01) | 7.52E+01(3.07E+00) | 2.54E+02(2.65E+01) |
| F19 | **8.19E+00(8.19E+00)** | 1.05E+03(3.85E+01) | 7.85E+03(6.28E+02) | 1.45E+03(1.78E+02) | 1.58E+01(5.79E-01) | 1.11E+03(2.97E+01) | 2.92E+03(4.50E+01) |
| F20 | **-2.65E-01(-2.65E-01)** | 1.44E+06(5.12E+04) | 5.87E+06(4.81E+05) | 2.58E+06(1.77E+05) | 3.08E+02(8.15E+01) | 1.25E+06(5.29E+04) | 3.17E+06(9.01E+04) |
| F21 | 8.04E+01(8.04E+01) | 9.09E+01(8.90E-01) | 4.74E+02(7.62E+01) | 1.11E+02(8.81E+00) | **7.60E+01(7.10E-01)** | 8.60E+01(3.71E-02) | 9.86E+01(1.34E+00) |
| F22 | 8.08E+01(8.08E+01) | 9.28E+01(3.31E+00) | 4.41E+02(4.83E+01) | 1.14E+02(8.09E+00) | **7.49E+01(2.79E+00)** | 8.62E+01(7.24E-02) | 9.76E+01(1.64E+00) |
| F23 | 1.68E+00(1.68E+00) | 1.05E+01(2.53E+00) | 2.93E+02(3.11E+01) | 3.82E+01(1.42E+01) | **1.65E+00(6.59E-02)** | 1.68E+00(2.10E-02) | 1.25E+01(1.24E+00) |
| F24 | 7.46E+03(7.46E+03) | 6.49E+05(1.51E+05) | 3.25E+07(4.36E+06) | 3.41E+06(6.66E+05) | **7.28E+03(7.25E+01)** | 2.21E+04(6.86E+02) | 1.25E+06(1.77E+05) |
| win/tie/loss | -/-/- | 22/1/1 | 23/0/1 | 23/0/1 | 18/2/4 | 22/1/1 | 23/0/1 |

## G Compare with TurBO

We compared POM and Bayesian optimization algorithms, finding that Bayesian optimization converges very slowly on high-dimensional problems. TurBO [56], noted for its fast convergence and strong performance [57], was used as a benchmark. Although TurBO requires substantial time for 10,000 evaluations, POM completes the same task in under one second. Therefore, we plotted the convergence curves of TurBO and POM with only 3,100 evaluations. As shown in Figure 14, POM demonstrates significant performance advantages over TurBO in most cases.

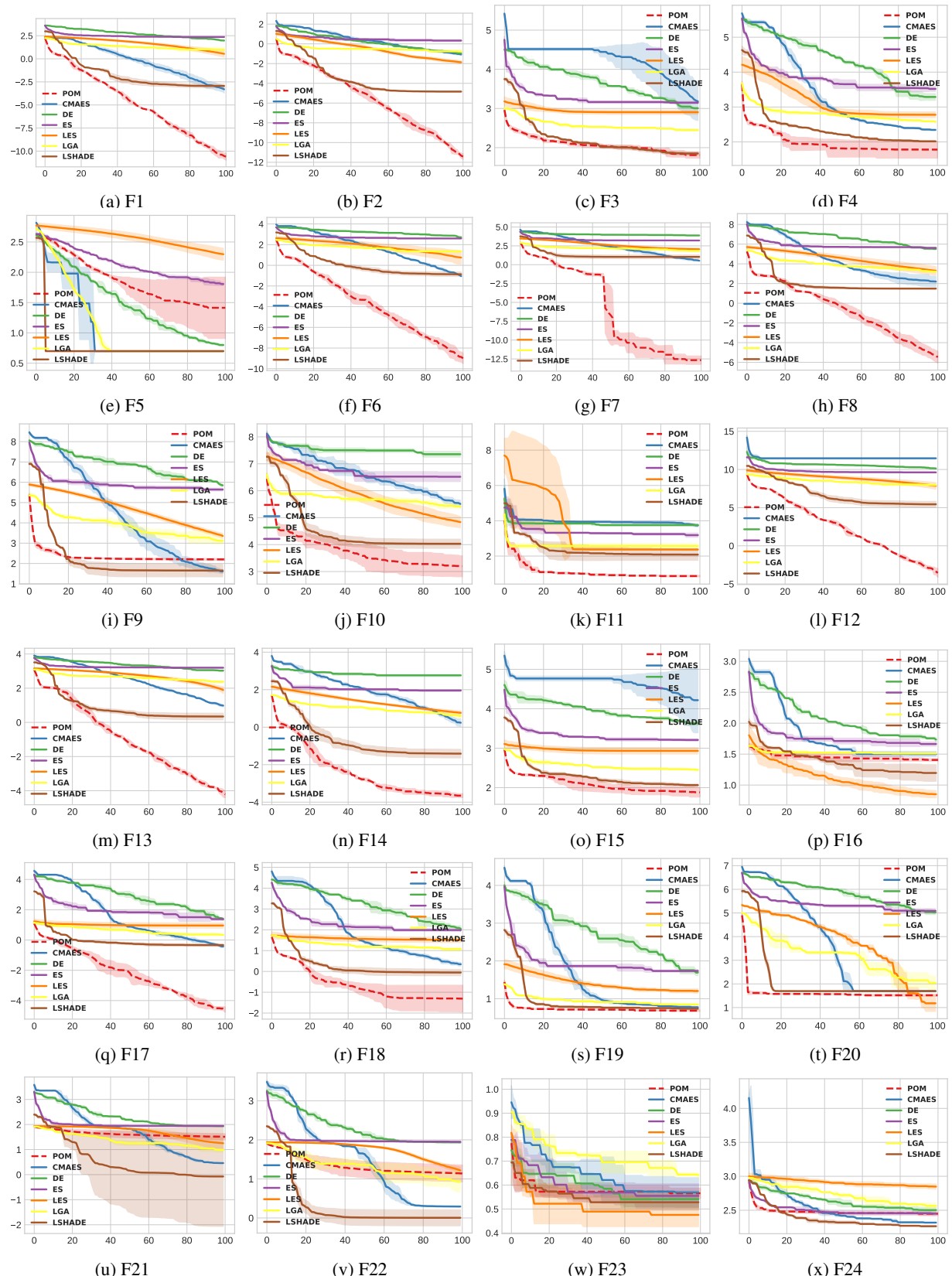

Figure 12: The log convergence curves of POM and other baselines. It shows the convergence curve of these algorithms on functions in BBOB with $d = 30$.

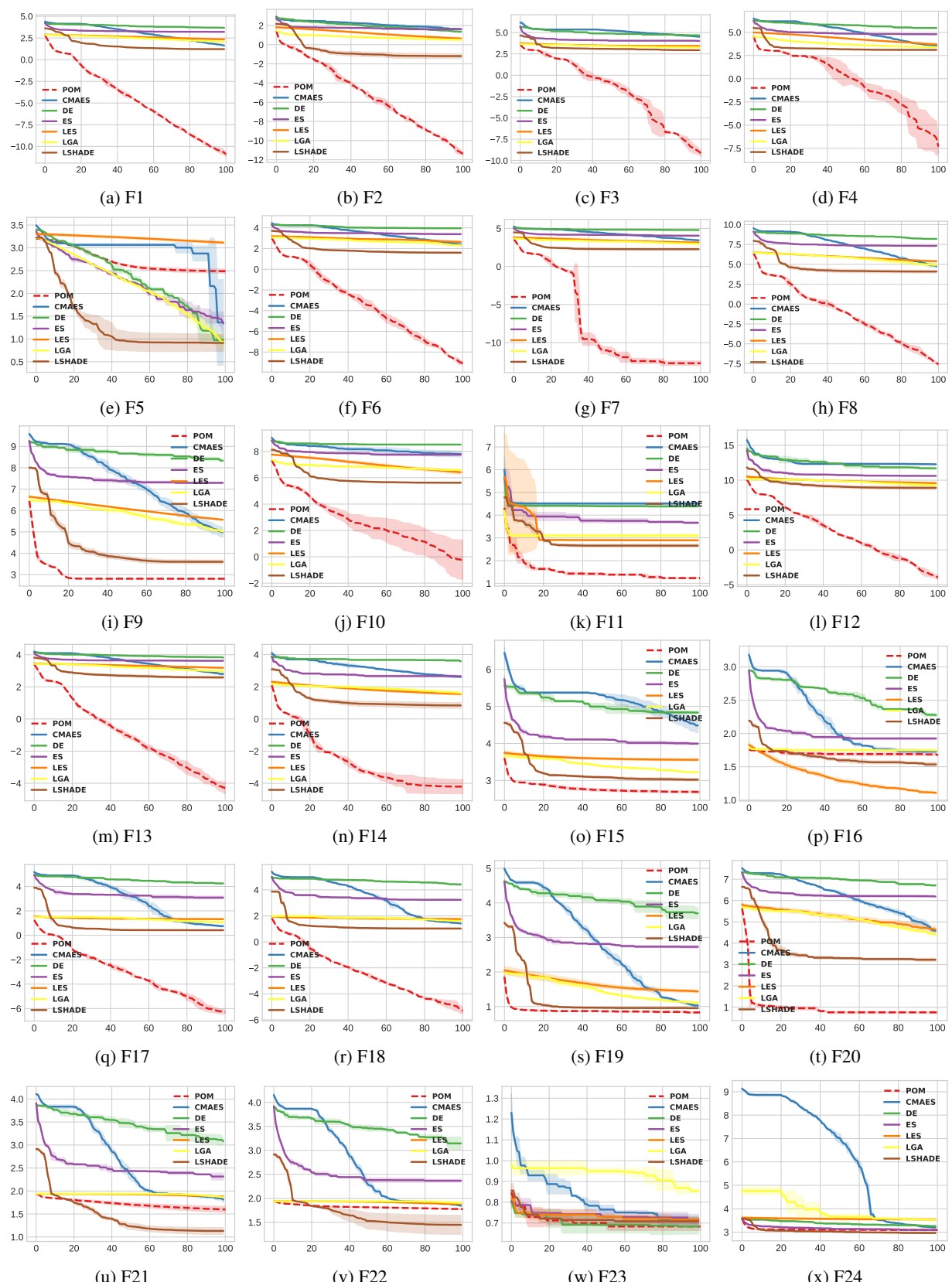

Figure 13: The log convergence curves of POM and other baselines. It shows the convergence curve of these algorithms on the functions in BBOB with $d = 100$.

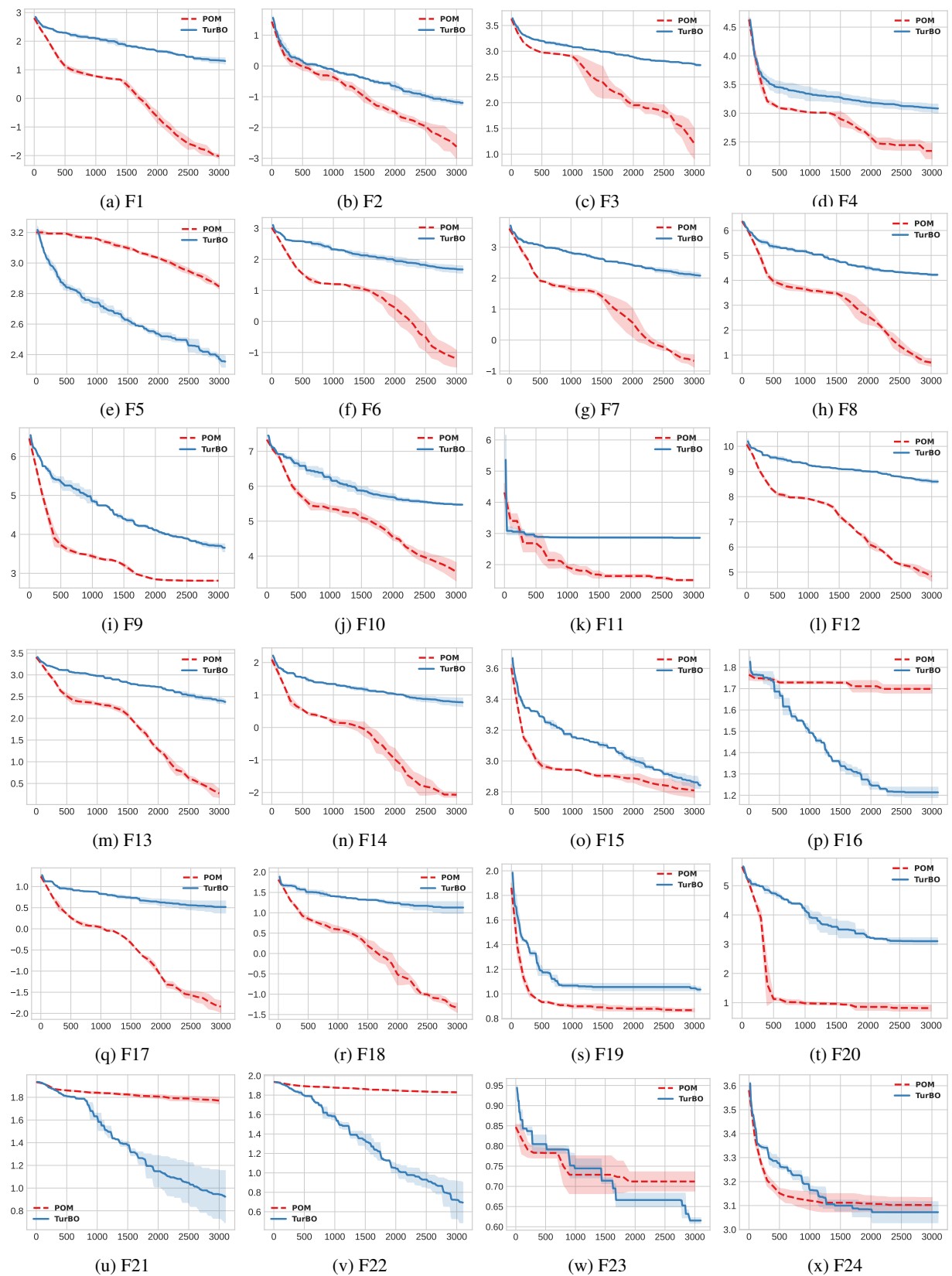

Figure 14: The log convergence curves of POM and TurBO. It shows the convergence curve of these algorithms on functions in BBOB with $d = 100$.

# H  Results of Analysis Study

Table 7: Results of ablation experiments. The best results are indicated in bold, and the suboptimal results are underlined. Here $d = 30$.

| F | NO LMM | NO LCM | NO MASK | UNTRAINED | POM |
|---|---|---|---|---|---|
| F1 | 8.85E+00(8.85E+00) | 5.86E-07(5.86E-07) | 3.09E+01(3.09E+01) | 4.74E-03(4.74E-03) | **5.58E-15(5.58E-15)** |
| F2 | 6.41E-03(6.41E-03) | 3.26E-08(3.26E-08) | 2.97E-01(2.97E-01) | 2.29E-06(2.29E-06) | **1.67E-17(1.67E-17)** |
| F3 | 2.77E+02(2.77E+02) | 5.55E+01(5.55E+01) | 4.34E+02(4.34E+02) | 1.56E+02(1.56E+02) | **1.34E+00(1.34E+00)** |
| F4 | 3.55E+02(3.55E+02) | 7.95E+01(7.95E+01) | 7.37E+02(7.37E+02) | 1.06E+03(1.06E+03) | **2.92E+01(2.92E+01)** |
| F5 | **2.39E+00(2.39E+00)** | 4.75E+01(4.75E+01) | 3.05E+01(3.05E+01) | 3.92E+01(3.92E+01) | 1.83E+01(1.83E+01) |
| F6 | 5.44E+01(5.44E+01) | 6.85E-06(6.85E-06) | 6.70E+01(6.70E+01) | 5.69E-02(5.69E-02) | **5.15E-13(5.15E-13)** |
| F7 | 3.30E+02(3.30E+02) | 2.51E-02(2.51E-02) | 1.70E+02(1.70E+02) | 1.71E+01(1.71E+01) | **6.28E-03(6.28E-03)** |
| F8 | 3.71E+03(3.71E+03) | 4.26E-04(4.26E-04) | 1.34E+04(1.34E+04) | 5.52E+00(5.52E+00) | **1.24E-11(1.24E-11)** |
| F9 | 5.37E+03(5.37E+03) | 1.51E+02(1.51E+02) | 8.09E+03(8.09E+03) | 1.36E+02(1.36E+02) | **1.32E+02(1.32E+02)** |
| F10 | 9.45E+05(9.45E+05) | **9.15E+02(9.15E+02)** | 6.47E+05(6.47E+05) | 2.73E+05(2.73E+05) | 1.22E+03(1.22E+03) |
| F11 | 2.69E+02(2.69E+02) | **9.26E+00(9.26E+00)** | 1.79E+02(1.79E+02) | 8.17E+01(8.17E+01) | 4.92E+01(4.92E+01) |
| F12 | 1.81E+08(1.81E+08) | 3.35E-02(3.35E-02) | 3.31E+08(3.31E+08) | 2.07E+04(2.07E+04) | **2.87E-07(2.87E-07)** |
| F13 | 2.95E+02(2.95E+02) | 2.76E-03(2.76E-03) | 5.71E+02(5.71E+02) | 1.76E+01(1.76E+01) | **1.05E-05(1.05E-05)** |
| F14 | 1.53E+01(1.53E+01) | 3.95E-04(3.95E-04) | 5.66E+00(5.66E+00) | 4.46E-02(4.46E-02) | **1.16E-04(1.16E-04)** |
| F15 | 3.71E+02(3.71E+02) | 1.14E+02(1.14E+02) | 3.84E+02(3.84E+02) | 1.94E+02(1.94E+02) | **9.35E+01(9.35E+01)** |
| F16 | 2.92E+01(2.92E+01) | 2.76E+01(2.76E+01) | 2.91E+01(2.91E+01) | 2.66E+01(2.66E+01) | **1.24E+01(1.24E+01)** |
| F17 | 6.19E+00(6.19E+00) | 2.23E-02(2.23E-02) | 4.75E+00(4.75E+00) | 2.97E-01(2.97E-01) | **5.06E-07(5.06E-07)** |
| F18 | 2.21E+01(2.21E+01) | 2.08E-01(2.08E-01) | 1.98E+00(1.98E+00) | 2.25E+00(2.25E+00) | **2.71E-03(2.71E-03)** |
| F19 | 8.12E+00(8.12E+00) | **5.06E+00(5.06E+00)** | 9.64E+00(9.64E+00) | 5.63E+00(5.63E+00) | 5.62E+00(5.62E+00) |
| F20 | 4.05E+02(4.05E+02) | **-2.75E+01(-2.75E+01)** | 1.56E+03(1.56E+03) | -3.11E+00(-3.11E+00) | -1.84E+01(-1.84E+01) |
| F21 | 3.11E+01(3.11E+01) | **2.20E+01(2.20E+01)** | 6.55E+01(6.55E+01) | 6.45E+01(6.45E+01) | 3.65E+01(3.65E+01) |
| F22 | **6.31E+00(6.31E+00)** | 1.08E+01(1.08E+01) | 5.10E+01(5.10E+01) | 6.32E+01(6.32E+01) | 3.86E+01(3.86E+01) |
| F23 | 3.63E+00(3.63E+00) | 3.49E+00(3.49E+00) | 3.31E+00(3.31E+00) | **3.23E+00(3.23E+00)** | 3.77E+00(3.77E+00) |
| F24 | 3.55E+02(3.55E+02) | **2.60E+02(2.60E+02)** | 3.98E+02(3.98E+02) | 2.88E+02(2.88E+02) | 3.41E+02(3.41E+02) |

Table 8: Results of Training Dataset Experiments. The best results are indicated in bold, and the suboptimal results are underlined.

| F | 1 | 2 | 3 | 4 | 5 | 6 | 7 | 8 |
|---|---|---|---|---|---|---|---|---|
| F1 | 1.50E-01(1.50E-01) | 5.33E-04(5.33E-04) | 8.26E-07(8.26E-07) | 5.56E-08(5.56E-08) | **4.97E-08(4.97E-08)** | 7.57E-06(7.57E-06) | 7.54E-05(7.54E-05) | 2.08E-04(2.08E-04) |
| F2 | 3.29E-02(3.29E-02) | 4.29E-05(4.29E-05) | 5.56E-07(5.56E-07) | 6.78E-07(6.78E-07) | **1.38E-09(1.38E-09)** | 6.62E-04(6.62E-04) | 6.41E-07(6.41E-07) | 6.54E-07(6.54E-07) |
| F3 | 8.52E+02(8.52E+02) | 5.40E+02(5.40E+02) | 2.87E+01(2.87E+01) | 5.61E+02(5.61E+02) | 3.84E+01(3.84E+01) | **8.46E+00(8.46E+00)** | 5.27E+02(5.27E+02) | 9.47E+01(9.47E+01) |
| F4 | 7.81E+02(7.81E+02) | 6.80E+02(6.80E+02) | 4.06E+02(4.06E+02) | 7.58E+02(7.58E+02) | 7.53E+02(7.53E+02) | 5.80E+02(5.80E+02) | 6.04E+02(6.04E+02) | **1.94E+02(1.94E+02)** |
| F5 | 9.14E+02(9.14E+02) | 1.14E+03(1.14E+03) | 7.18E+02(7.18E+02) | 7.74E+02(7.74E+02) | **5.48E+02(5.48E+02)** | 7.04E+02(7.04E+02) | 5.93E+02(5.93E+02) | 7.66E+02(7.66E+02) |
| F6 | 9.30E-01(9.30E-01) | 2.89E-03(2.89E-03) | 1.11E-05(1.11E-05) | 1.04E-05(1.04E-05) | **1.09E-06(1.09E-06)** | 7.17E-05(7.17E-05) | 6.30E-04(6.30E-04) | 8.35E-04(8.35E-04) |
| F7 | 1.06E+00(1.06E+00) | 2.00E-11(2.00E-11) | 1.60E-12(1.60E-12) | 2.98E-12(2.98E-12) | **1.40E-13(1.40E-13)** | 1.09E-12(1.09E-12) | 7.43E-12(7.43E-12) | 1.18E-11(1.18E-11) |
| F8 | 4.94E+00(4.94E+00) | 1.13E+00(1.13E+00) | 7.35E-03(7.35E-03) | **2.97E-04(2.97E-04)** | 4.82E+01(4.82E+01) | 5.00E+00(5.00E+00) | 5.27E-02(5.27E-02) | 5.92E-02(5.92E-02) |
| F9 | 1.56E+03(1.56E+03) | 6.43E+02(6.43E+02) | **6.36E+02(6.36E+02)** | 6.41E+02(6.41E+02) | 6.40E+02(6.40E+02) | 6.42E+02(6.42E+02) | 6.41E+02(6.41E+02) | 6.39E+02(6.39E+02) |
| F10 | 2.39E+04(2.39E+04) | 3.60E+02(3.60E+02) | 1.17E-01(1.17E-01) | 3.28E-01(3.28E-01) | **1.65E-04(1.65E-04)** | 3.32E+01(3.32E+01) | 3.32E+00(3.32E+00) | 1.83E+00(1.83E+00) |
| F11 | 2.12E+02(2.12E+02) | 2.04E+01(2.04E+01) | 1.58E+01(1.58E+01) | 2.86E+01(2.86E+01) | 1.97E+01(1.97E+01) | 2.91E+01(2.91E+01) | 1.23E+01(1.23E+01) | **9.63E+00(9.63E+00)** |
| F12 | 3.60E+07(3.60E+07) | 1.51E+04(1.51E+04) | 8.00E+00(8.00E+00) | 2.00E+01(2.00E+01) | **8.97E-01(8.97E-01)** | 2.52E+03(2.52E+03) | 1.00E+03(1.00E+03) | 3.18E+02(3.18E+02) |
| F13 | 2.08E+01(2.08E+01) | 1.67E+00(1.67E+00) | 5.91E-02(5.91E-02) | **7.06E-03(7.06E-03)** | 3.42E-02(3.42E-02) | 2.63E-01(2.63E-01) | 8.87E-01(8.87E-01) | 1.51E+00(1.51E+00) |
| F14 | 1.24E-01(1.24E-01) | 4.11E-03(4.11E-03) | 9.29E-05(9.29E-05) | 1.37E-05(1.37E-05) | **5.94E-06(5.94E-06)** | 1.24E-03(1.24E-03) | 1.37E-03(1.37E-03) | 1.75E-03(1.75E-03) |
| F15 | 8.38E+02(8.38E+02) | **6.47E+02(6.47E+02)** | 6.55E+02(6.55E+02) | 7.32E+02(7.32E+02) | 6.48E+02(6.48E+02) | 7.21E+02(7.21E+02) | 7.90E+02(7.90E+02) | 7.66E+02(7.66E+02) |
| F16 | 5.01E+01(5.01E+01) | 4.77E+01(4.77E+01) | 4.70E+01(4.70E+01) | **4.65E+01(4.65E+01)** | **4.65E+01(4.65E+01)** | 4.84E+01(4.84E+01) | 4.72E+01(4.72E+01) | 4.74E+01(4.74E+01) |
| F17 | 6.55E-01(6.55E-01) | 1.02E-01(1.02E-01) | 3.08E-02(3.08E-02) | 1.47E-02(1.47E-02) | **5.44E-04(5.44E-04)** | 3.90E-03(3.90E-03) | 3.15E-02(3.15E-02) | 6.75E-02(6.75E-02) |
| F18 | 2.28E+00(2.28E+00) | 4.04E-01(4.04E-01) | 7.99E-02(7.99E-02) | 1.47E-01(1.47E-01) | **3.25E-03(3.25E-03)** | 4.09E-02(4.09E-02) | 1.28E-01(1.28E-01) | 2.79E-01(2.79E-01) |
| F19 | 7.88E+00(7.88E+00) | 7.28E+00(7.28E+00) | 7.04E+00(7.04E+00) | 7.07E+00(7.07E+00) | **6.44E+00(6.44E+00)** | 6.57E+00(6.57E+00) | 7.26E+00(7.26E+00) | 7.28E+00(7.28E+00) |
| F20 | 2.60E-01(2.60E-01) | 1.33E+00(1.33E+00) | 2.07E+00(2.07E+00) | **-1.54E+00(-1.54E+00)** | 0(0) | 6.32E-01(6.32E-01) | 1.95E+00(1.95E+00) | 2.00E+00(2.00E+00) |
| F21 | 6.45E+01(6.45E+01) | 6.87E+01(6.87E+01) | **5.72E+01(5.72E+01)** | 6.67E+01(6.67E+01) | 5.94E+01(5.94E+01) | 6.72E+01(6.72E+01) | 6.09E+01(6.09E+01) | 6.36E+01(6.36E+01) |
| F22 | 7.55E+01(7.55E+01) | 7.65E+01(7.65E+01) | 7.62E+01(7.62E+01) | 7.73E+01(7.73E+01) | **7.38E+01(7.38E+01)** | 7.69E+01(7.69E+01) | 7.62E+01(7.62E+01) | 7.46E+01(7.46E+01) |
| F23 | 5.21E+00(5.21E+00) | **4.74E+00(4.74E+00)** | 5.20E+00(5.20E+00) | 5.21E+00(5.21E+00) | 5.19E+00(5.19E+00) | 4.92E+00(4.92E+00) | 5.08E+00(5.08E+00) | 4.83E+00(4.83E+00) |
| F24 | 1.33E+03(1.33E+03) | 1.41E+03(1.41E+03) | 1.24E+03(1.24E+03) | 1.26E+03(1.26E+03) | 1.22E+03(1.22E+03) | 1.31E+03(1.31E+03) | **1.19E+03(1.19E+03)** | 1.31E+03(1.31E+03) |

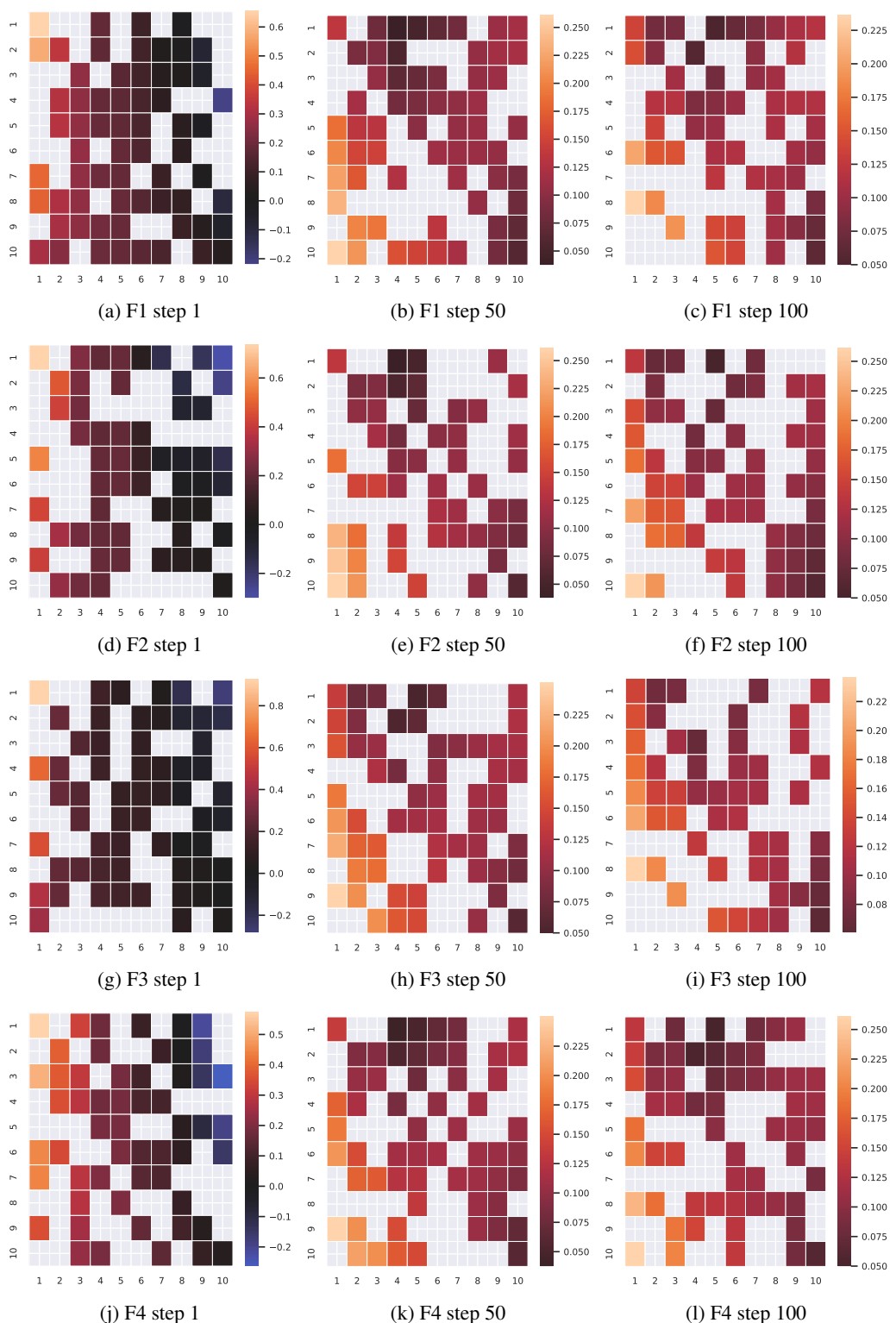

Figure 15: Visualized results of mutation strategy $S^t$ on BBOB (F1-F4) with $d = 100$. Here, $n = 10$ for the sake of clarity. The blank squares in the matrix indicate the masked parts in Eq. (8). Steps 1, 50 and 100 correspond to the 1st, 50th and 100th generations in the population evolution process. The horizontal and vertical axes show the ranking of individuals, with 1 being the best and 10 being the worst in the population. Each row represents the weight assigned to other individuals when performing mutation operations for the corresponding individual.

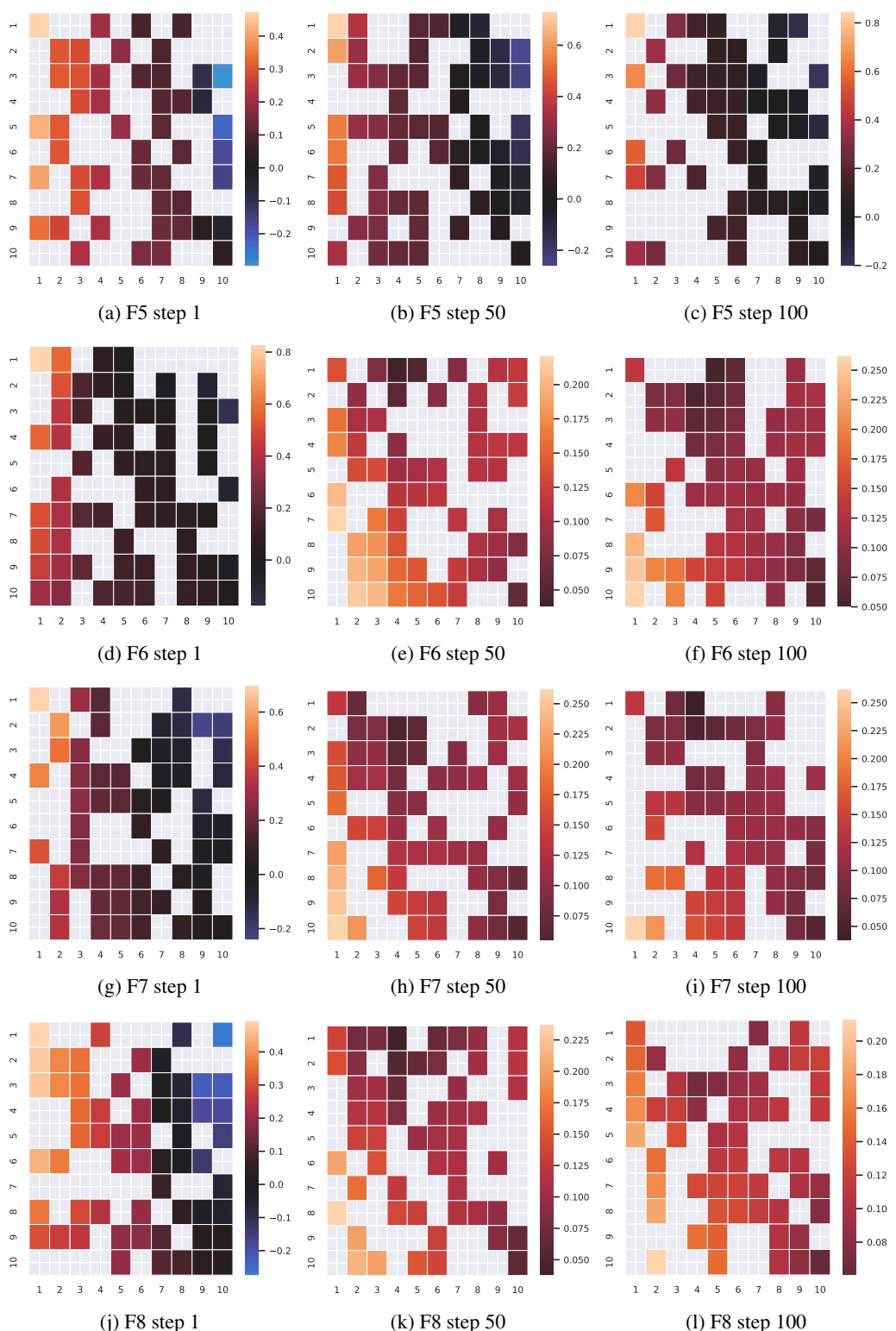

Figure 16: Visualized results of mutation strategy $S^t$ on BBOB (F5-F8) with $d = 100$.

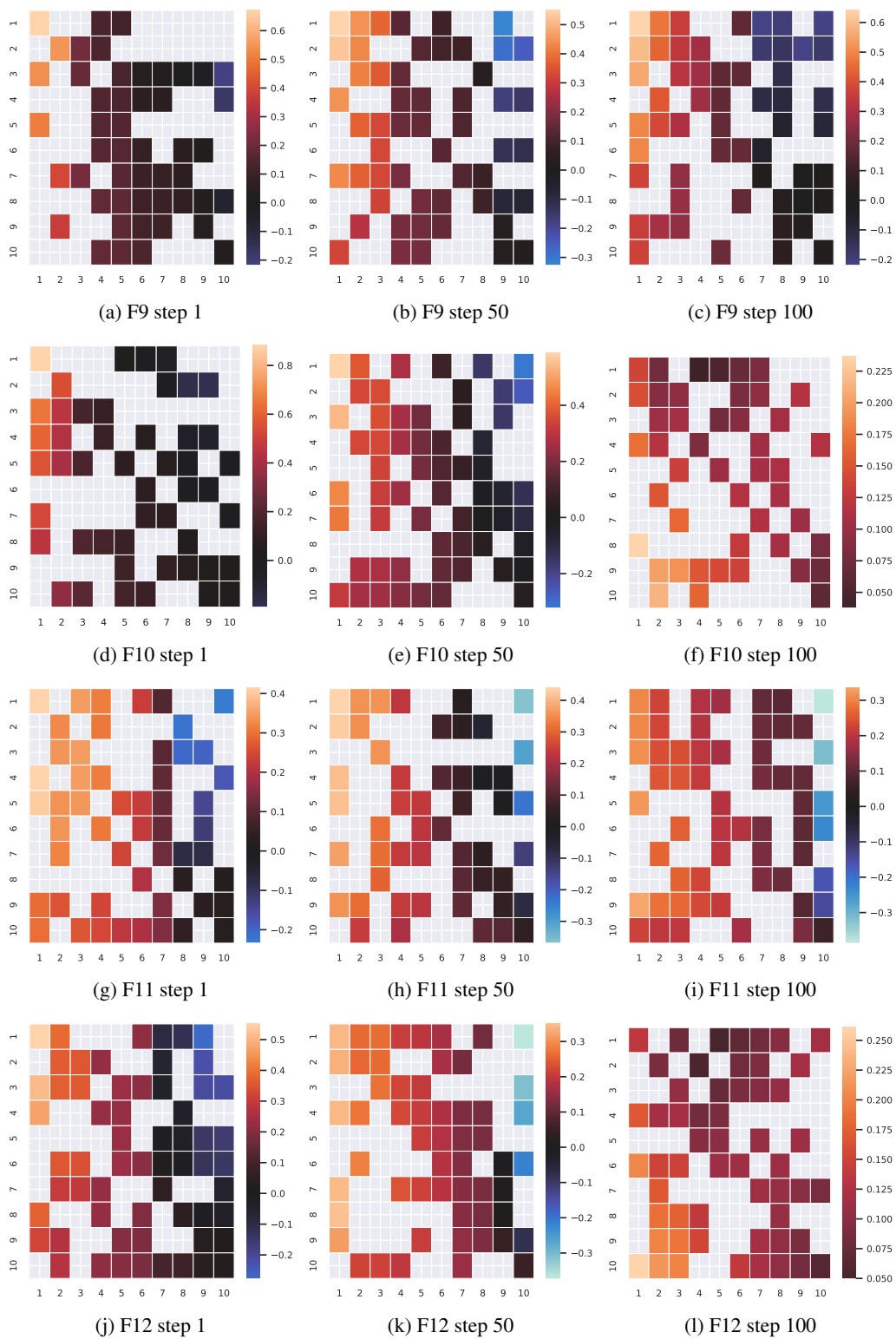

(a) F9 step 1     (b) F9 step 50     (c) F9 step 100

(d) F10 step 1     (e) F10 step 50     (f) F10 step 100

(g) F11 step 1     (h) F11 step 50     (i) F11 step 100

(j) F12 step 1     (k) F12 step 50     (l) F12 step 100

Figure 17: Visualized results of mutation strategy $S^t$ on BBOB (F9-F12) with $d = 100$.

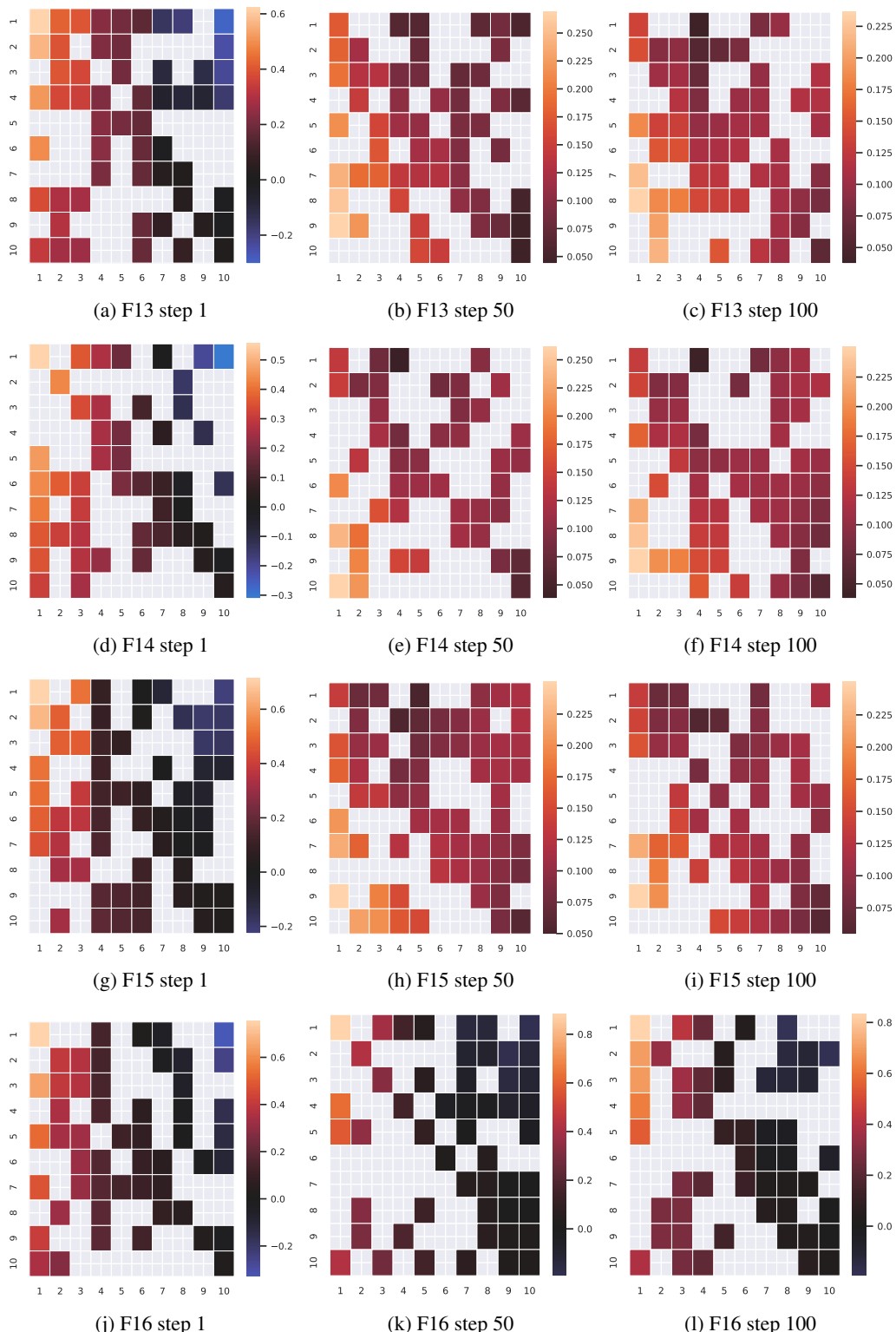

Figure 18: Visualized results of mutation strategy $S^t$ on BBOB (F13-F16) with $d = 100$.

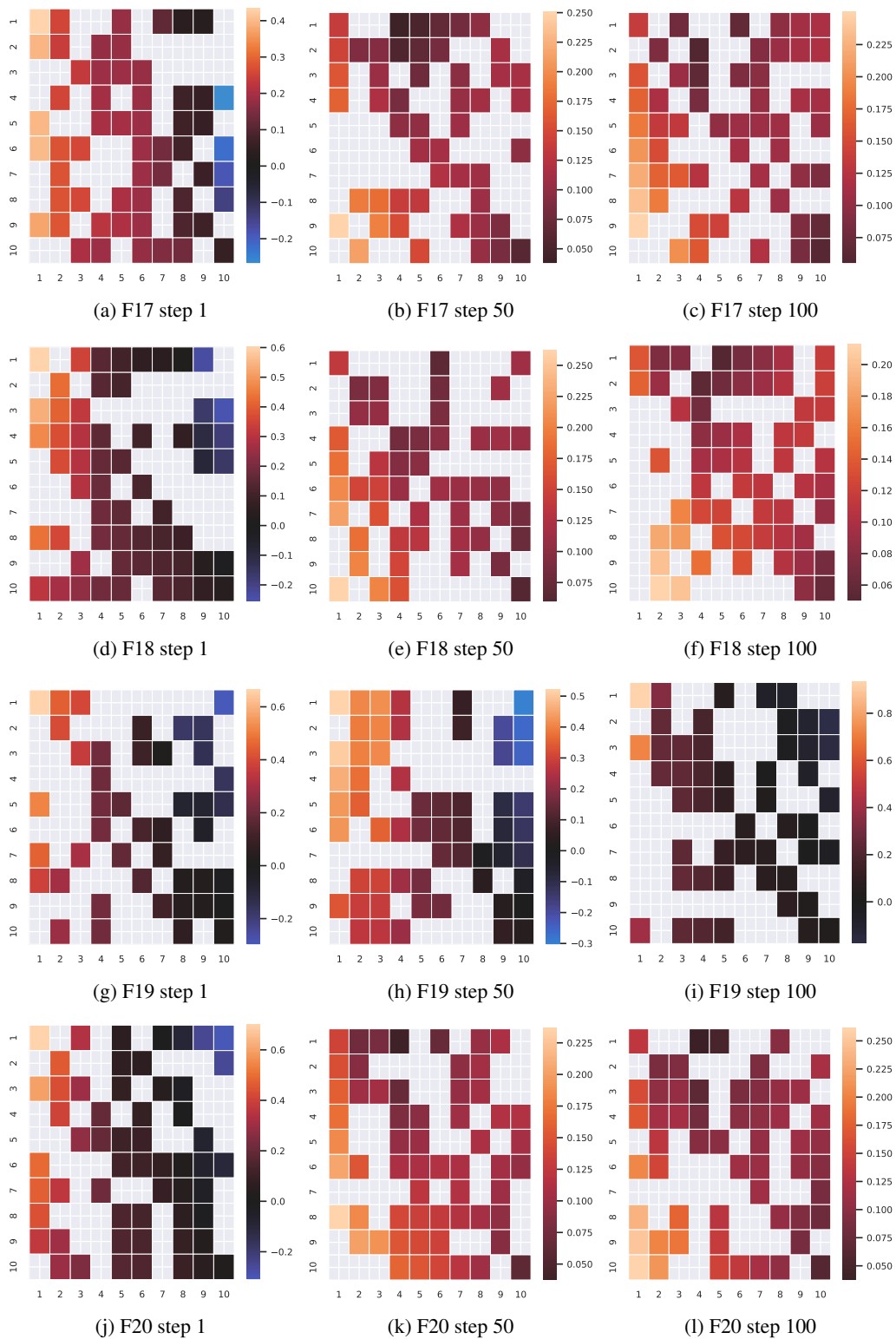

Figure 19: Visualized results of mutation strategy $S^t$ on BBOB (F17-F20) with $d = 100$.

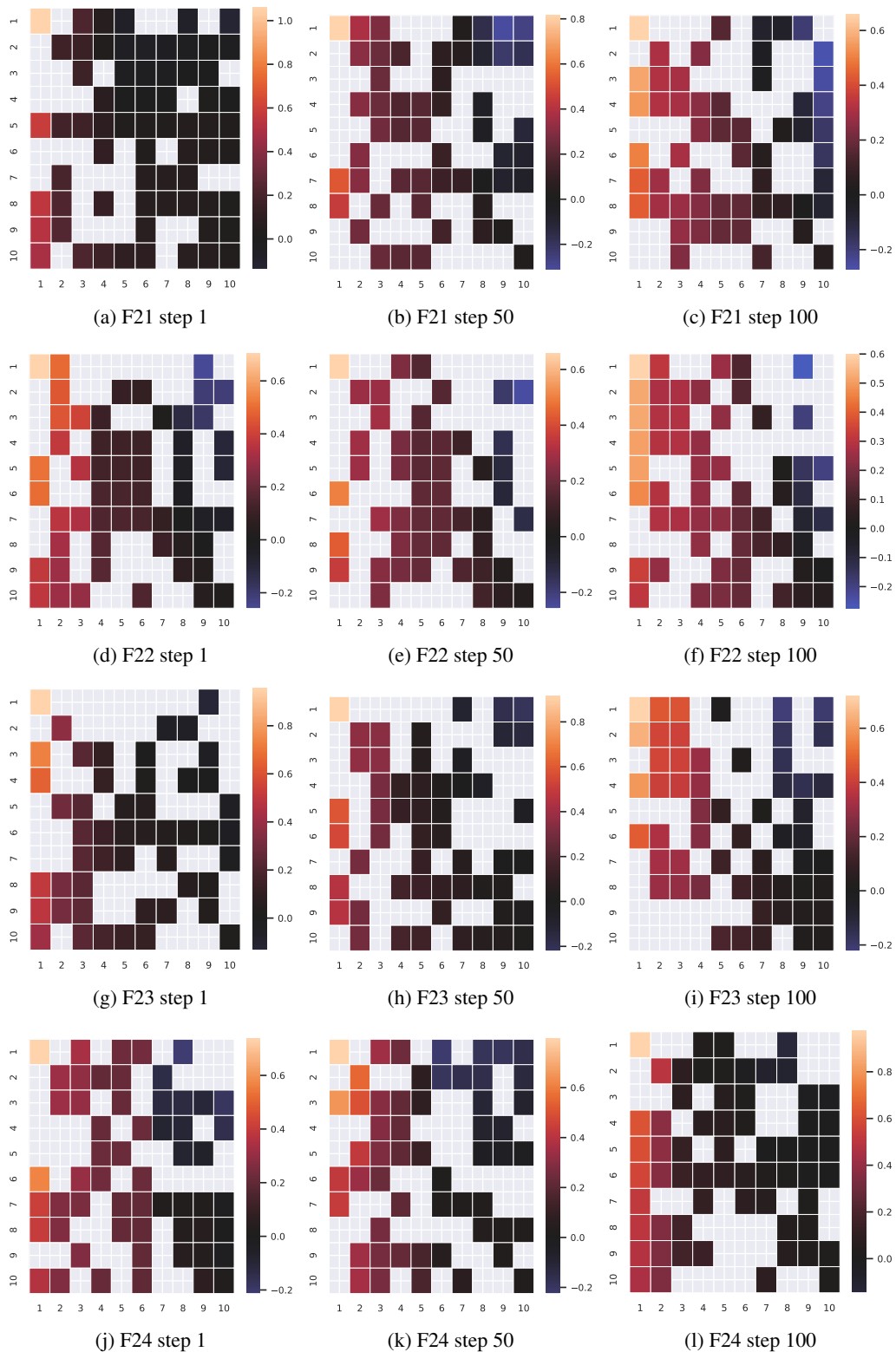

(a) F21 step 1          (b) F21 step 50          (c) F21 step 100

(d) F22 step 1          (e) F22 step 50          (f) F22 step 100

(g) F23 step 1          (h) F23 step 50          (i) F23 step 100

(j) F24 step 1          (k) F24 step 50          (l) F24 step 100

Figure 20: Visualized results of mutation strategy $S^t$ on BBOB (F21-F24) with $d = 100$.

Table 9: Results of POMs of different sizes on BBOB tests ($d = 100$). The best results are indicated in bold, and the suboptimal results are underlined.

| F | XS | S | M | L | VL | XL |
|---|---|---|---|---|---|---|
| F1 | 3.89E-16(3.89E-16) | 5.38E-07(5.38E-07) | 6.06E-19(6.06E-19) | 7.88E-24(7.88E-24) | 1.97E-19(1.97E-19) | **4.99E-27(4.99E-27)** |
| F2 | 7.66E-18(7.66E-18) | 9.97E-09(9.97E-09) | 2.43E-21(2.43E-21) | **6.96E-30(6.96E-30)** | 1.04E-21(1.04E-21) | 2.58E-29(2.58E-29) |
| F3 | 3.11E-16(3.11E-16) | 4.60E+02(4.60E+02) | 1.39E-17(1.39E-17) | 2.89E-16(2.89E-16) | 4.17E+02(4.17E+02) | **3.34E-26(3.34E-26)** |
| F4 | 8.49E-03(8.49E-03) | 5.26E+02(5.26E+02) | 2.46E-02(2.46E-02) | 8.56E-01(8.56E-01) | 3.10E+02(3.10E+02) | **6.15E-23(6.15E-23)** |
| F5 | 4.00E+02(4.00E+02) | 3.22E+02(3.22E+02) | 3.89E+02(3.89E+02) | **0.00E+00(0.00E+00)** | 4.05E+02(4.05E+02) | 2.56E+02(2.56E+02) |
| F6 | 2.94E-14(2.94E-14) | 8.11E-06(8.11E-06) | 1.58E-16(1.58E-16) | 1.67E-19(1.67E-19) | 6.81E-17(6.81E-17) | **3.92E-24(3.92E-24)** |
| F7 | **1.83E-14(1.83E-14)** | 4.03E-13(4.03E-13) | 9.48E-14(9.48E-14) | 2.22E-13(2.22E-13) | 6.61E-14(6.61E-14) | 9.54E-14(9.54E-14) |
| F8 | **0.00E+00(0.00E+00)** | 6.76E-04(6.76E-04) | 0.00E+00(0.00E+00) | 0.00E+00(0.00E+00) | 0.00E+00(0.00E+00) | 0.00E+00(0.00E+00) |
| F9 | 6.32E+02(6.32E+02) | 6.42E+02(6.42E+02) | 6.37E+02(6.37E+02) | **6.27E+02(6.27E+02)** | 6.38E+02(6.38E+02) | 6.38E+02(6.38E+02) |
| F10 | 6.86E-01(6.86E-01) | 8.96E+02(8.96E+02) | 9.37E+01(9.37E+01) | 1.89E+04(1.89E+04) | 1.26E+00(1.26E+00) | **2.45E-02(2.45E-02)** |
| F11 | 2.93E+01(2.93E+01) | 1.34E+02(1.34E+02) | 8.50E+01(8.50E+01) | 6.84E+01(6.84E+01) | 4.56E+01(4.56E+01) | **1.10E+01(1.10E+01)** |
| F12 | 2.84E-11(2.84E-11) | 2.90E+00(2.90E+00) | 6.53E-11(6.53E-11) | 1.02E-03(1.02E-03) | 2.11E-09(2.11E-09) | **3.89E-20(3.89E-20)** |
| F13 | 1.17E-07(1.17E-07) | 3.47E-02(3.47E-02) | 1.95E-07(1.95E-07) | 1.29E-05(1.29E-05) | 1.67E-06(1.67E-06) | **1.08E-11(1.08E-11)** |
| F14 | 2.85E-05(2.85E-05) | 1.97E-04(1.97E-04) | 1.71E-05(1.71E-05) | 8.94E-05(8.94E-05) | 1.00E-04(1.00E-04) | **6.58E-06(6.58E-06)** |
| F15 | 9.76E+01(9.76E+01) | 5.18E+02(5.18E+02) | 3.22E+02(3.22E+02) | 5.57E+02(5.57E+02) | 4.95E+02(4.95E+02) | **1.27E-07(1.27E-07)** |
| F16 | 3.45E+01(3.45E+01) | 4.51E+01(4.51E+01) | **2.65E+01(2.65E+01)** | 3.34E+01(3.34E+01) | 3.13E+01(3.13E+01) | 3.50E+01(3.50E+01) |
| F17 | 1.63E-08(1.63E-08) | 1.15E-03(1.15E-03) | 5.86E-10(5.86E-10) | 6.79E-11(6.79E-11) | 6.60E-10(6.60E-10) | **1.61E-14(1.61E-14)** |
| F18 | 3.12E-08(3.12E-08) | 5.33E-03(5.33E-03) | 5.16E-09(5.16E-09) | 4.85E-08(4.85E-08) | 1.61E-08(1.61E-08) | **5.50E-14(5.50E-14)** |
| F19 | 6.22E+00(6.22E+00) | 7.26E+00(7.26E+00) | 6.59E+00(6.59E+00) | 7.06E+00(7.06E+00) | 7.34E+00(7.34E+00) | **6.21E+00(6.21E+00)** |
| F20 | -5.39E+00(-5.39E+00) | -7.19E+00(-7.19E+00) | **-7.71E+00(-7.71E+00)** | 9.90E-01(9.90E-01) | -4.40E+00(-4.40E+00) | -1.94E+00(-1.94E+00) |
| F21 | 5.24E+01(5.24E+01) | 6.86E+01(6.86E+01) | 5.46E+01(5.46E+01) | **2.22E+01(2.22E+01)** | 6.43E+01(6.43E+01) | 4.88E+01(4.88E+01) |
| F22 | 7.24E+01(7.24E+01) | 7.70E+01(7.70E+01) | 7.30E+01(7.30E+01) | **6.73E+01(6.73E+01)** | 7.65E+01(7.65E+01) | 7.34E+01(7.34E+01) |
| F23 | 5.06E+00(5.06E+00) | 5.17E+00(5.17E+00) | 5.15E+00(5.15E+00) | 5.17E+00(5.17E+00) | 5.42E+00(5.42E+00) | **5.04E+00(5.04E+00)** |
| F24 | **1.31E+03(1.31E+03)** | 1.37E+03(1.37E+03) | 1.31E+03(1.31E+03) | 1.32E+03(1.32E+03) | 1.35E+03(1.35E+03) | 1.34E+03(1.34E+03) |

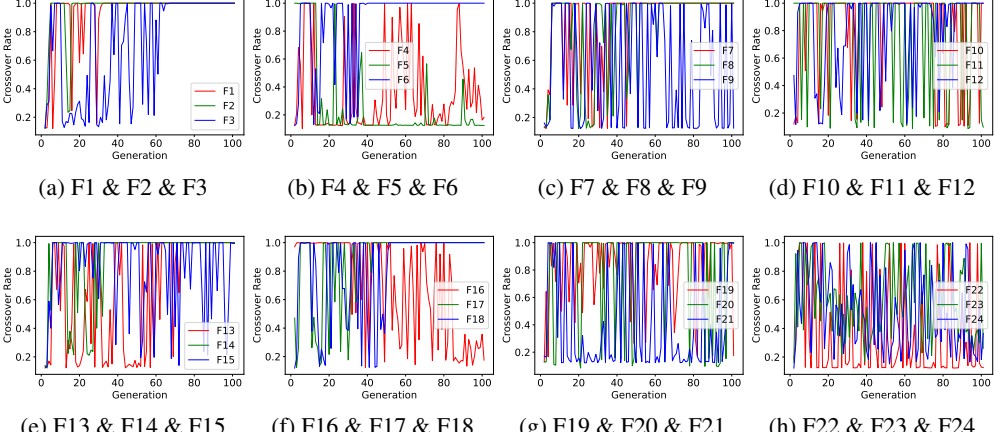

(a) F1 & F2 & F3  (b) F4 & F5 & F6  (c) F7 & F8 & F9  (d) F10 & F11 & F12

(e) F13 & F14 & F15  (f) F16 & F17 & F18  (g) F19 & F20 & F21  (h) F22 & F23 & F24

Figure 21: Results of a visual analysis of LCM on BBOB with $d = 100$. Here, $n = 100$. This is the crossover strategy of the individual ranked No. 1. Rank denotes the ranking of an individual. A subgraph illustrates the change in the probability of an individual crossing three tasks as the population evolves.

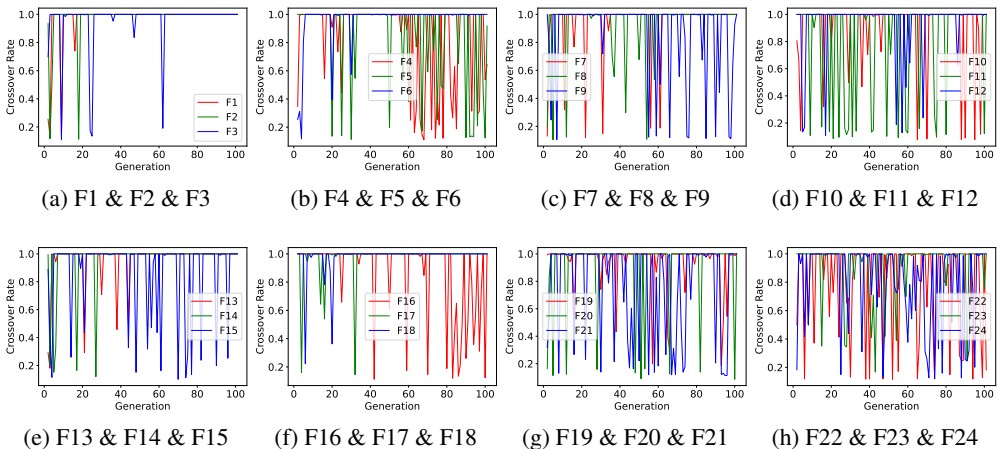

Figure 22: Results of a visual analysis of LCM on BBOB with $d = 100$. Here, $n = 100$. This is the crossover strategy of the individual ranked No. 5. Rank denotes the ranking of an individual. A subgraph illustrates the change in the probability of an individual crossing three tasks as the population evolves.

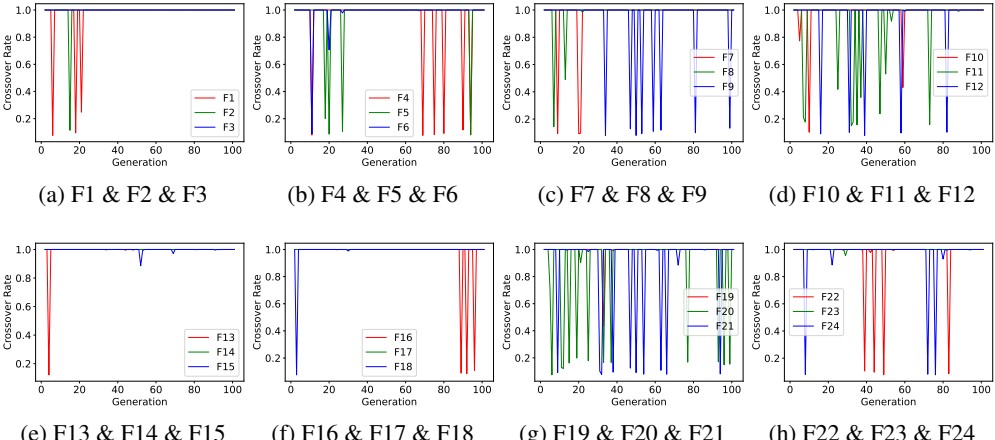

Figure 23: Results of a visual analysis of LCM on BBOB with $d = 100$. Here, $n = 100$. This is the crossover strategy of the individual ranked No. 18. Rank denotes the ranking of an individual. A subgraph illustrates the change in the probability of an individual crossing three tasks as the population evolves.

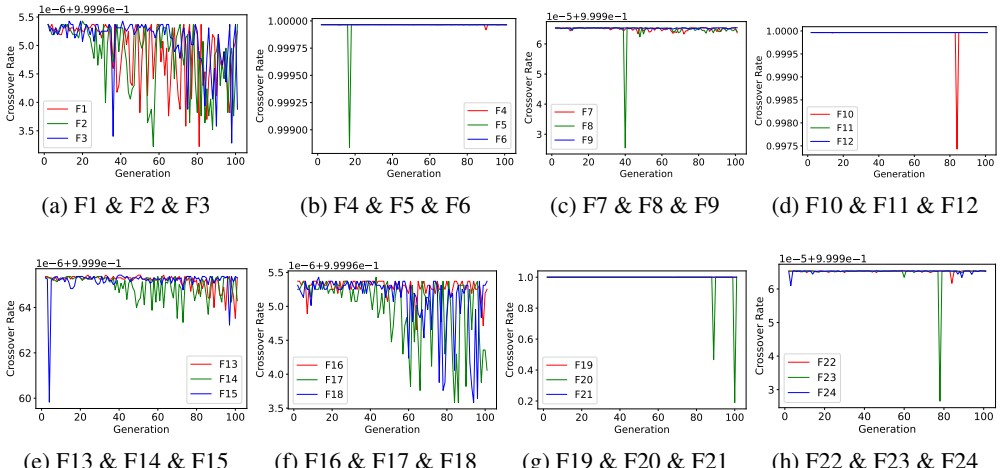

Figure 24: Results of a visual analysis of LCM on BBOB with $d = 100$. Here, $n = 100$. This is the crossover strategy of the individual ranked No. 51. Rank denotes the ranking of an individual. A subgraph illustrates the change in the probability of an individual crossing three tasks as the population evolves.

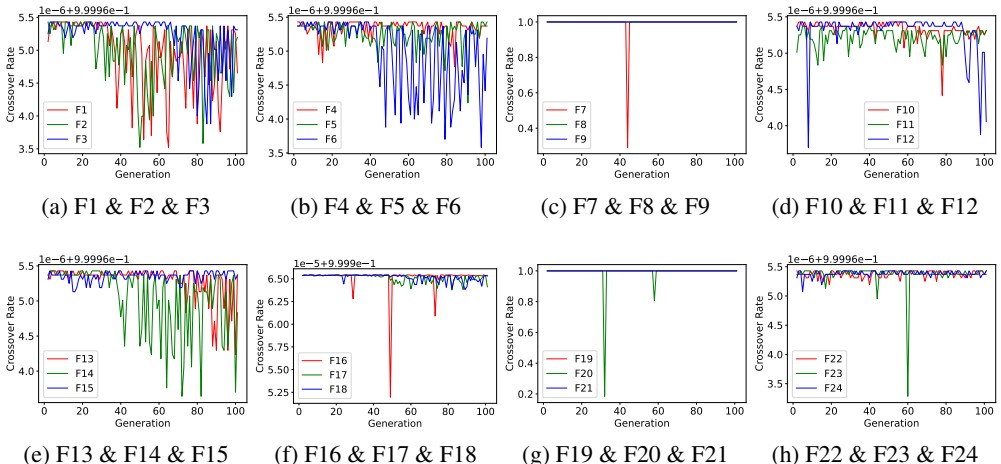

Figure 25: Results of a visual analysis of LCM on BBOB with $d = 100$. Here, $n = 100$. This is the crossover strategy of the individual ranked No. 75. Rank denotes the ranking of an individual. A subgraph illustrates the change in the probability of an individual crossing three tasks as the population evolves.

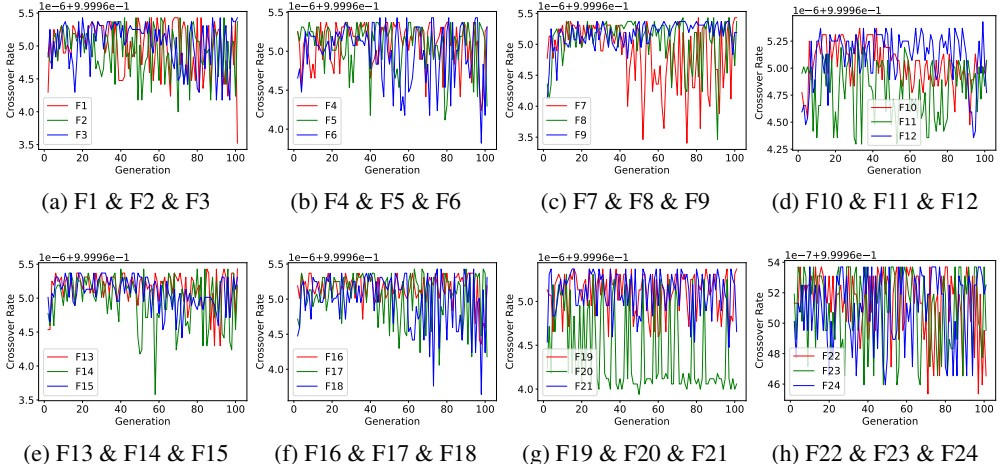

(a) F1 & F2 & F3      (b) F4 & F5 & F6      (c) F7 & F8 & F9      (d) F10 & F11 & F12

(e) F13 & F14 & F15   (f) F16 & F17 & F18   (g) F19 & F20 & F21   (h) F22 & F23 & F24

Figure 26: Results of a visual analysis of LCM on BBOB with $d = 100$. Here, $n = 100$. This is the crossover strategy of the individual ranked No. 100. Rank denotes the ranking of an individual. A subgraph illustrates the change in the probability of an individual crossing three tasks as the population evolves.

## I  Limitations

- Model size: In the experiment, we found that the relationship between the model size and the performance of POM is not a strict linear relationship. Although the larger the model, the more difficult it is to train, there is still no very quantitative design criterion between model size, training data volume and training difficulty.

- Time performance: We introduced an operation similar to the attention mechanism, whose time complexity is $O(n^2)$, which makes POM require a lot of time cost when processing large-scale populations. How to reduce and improve the time efficiency of POM is also worthy of further study.

## J  Potential Impact

This paper presents work whose goal is to advance the field of Machine Learning. There are many potential societal consequences of our work, none which we feel must be specifically highlighted here.

