# OpenReview forum: "Pretrained Optimization Model for Zero-Shot Black Box Optimization"
_NeurIPS.cc/2024/Conference — NeurIPS 2024 poster_

### Official Review · Reviewer_Se7V · 2024-07-12

**Soundness:** 3
**Presentation:** 3
**Contribution:** 3
**Rating:** 7
**Confidence:** 5

**Summary:**

This paper focus on zero-shot black box optimization. They propose a pretrained optimization model (POM) to pretrain on a training dataset, achieving good results on BBOB benchmark and two robot control tasks. The authors design several parts including LMM and LCM for the POM to achieve the generation of sample strategy. The experiment results are comprehensive with the main experiment and some in-depth experiments.

**Strengths:**

1. The design of the POM make sense in general. The authors draw inspiration from optimization process and hand-crafted a lot of design for the LMM and LCM.
2. The training process make sense too. The construction of the training dataset consider the coverage of the landscape feature. The design of loss function is really instructive, considering both convergence and diversity.
3. The algorithm present good experiment results in both synthetic BBOB dataset and robot control task, and achieve good results on both low and high dimensional scenario.

**Weaknesses:**

1. POM contains so many hand-crafted design, but are still under the scheme of mutate and then crossover, what's the motivation to decide this is the best design?
2. From the 'Ablation Study', we learn that the `mask` is the most important design of the POM which is just a random process without any theoretical guarantee. This would raise the concern of the whole design of the framework because those hand-crafted parts aren't even more important than the random mask. Also the NOTRAIN one perform better when one of the core design been discarded, which is weird.
3. More results can be presented. (e.g. there are only results for 100D when BBOB test with optimal solution disturbed)
4. The training dataset is relatively simple, it's easy for a model to overfit to it.
5. Related work can be more comprehensive. For example, for the part of 'LLM for Optimization', there are several works working in resolving black-box optimization problems using LLMs, such as https://arxiv.org/abs/2403.01131, https://arxiv.org/abs/2401.02051...). It would be nice to have a more comprehensive introduction to these related works.

**Questions:**

1. What will the results be when functions in BBOB dataset being rotated and shifted simultaneously? In the main experiment, the optimal points are all located in 0. Although in appendix there are results for shifted version, but rotation is also another usual operation to examine the robustness of some algorithms.
2. How do you implement those baseline methods? It will be nice to mention this.
3. For LES and LGA, was them being trained in the same dataset under the setting in this paper?

**Limitations:**

Please see weaknees and question part.

---

> ### Author Rebuttal · Authors · 2024-08-07
>
> **weakness 1**
>
> We do not claim that the current version of POM is the best design, but it does show excellent performance in experiments. We also do not emphasize that POM must be under the scheme of mutate and then crossover. We just design LMM and LCM modules for solution generation according to the needs of POM, and ensure that POM can be pre-trained end-to-end through gradient.
>
> **weakness 2**
>
> The problems you mentioned just prove the rationality of the POM module design.
>
> 1) mask
>
> The reason for introducing `mask` is explained in detail in lines 121-126. Its function is to limit the information exchange within the population, prevent the population from quickly converging to the optimal individual, resulting in loss of diversity. This design is similar to the dropout operation of deep neural networks. Without the dropout operation, the neural network will easily fall into the local optimum and overfit on the training set.
>
> It is worth noting that the role of `mask` is to make LCM and LMM work better. `mask` is like a catalyst, while LCM and LMM can be likened to reactants. We cannot say that the catalyst itself is more important than the reactants.
>
> 2) `NOTTRAIN` and `NO LMM`
>
> `NOTRAIN` performs better than `NO LMM`, which means LMM is a very core component. If LMM is lost, the trained model performance is not as good as `NOTRAIN`.
>
> **weakness 3**
>
> The experimental results of BBOB (d=100 ,optimal solutions disturbed), Bipedal Walker and Enduro proved the excellent performance of POM.  These experimental results have been able to verify the optimization and generalization capabilities of POM. We further show the experimental results of BBOB(d=500, disturbed).
>
> |      **F**       |         **GLHF**         |       **CMA-ES**       |       **LSHADE**       |
> | :--------------: | :----------------------: | :--------------------: | :--------------------: |
> |      **F1**      |  **1.29E+02(1.29E+02**)  |   6.12E+02(3.15E+01)   |   2.06E+02(2.18E+01)   |
> |      **F2**      |    5.22E+00(5.22E+00)    |   6.28E+01(3.25E+00)   | **3.13E+00(5.25E-01)** |
> |      **F3**      |    4.55E+03(4.55E+03)    |   8.93E+03(2.57E+02)   |   4.00E+03(1.15E+02)   |
> |      **F4**      |    9.98E+03(9.98E+03)    |   2.62E+04(3.08E+03)   | **8.82E+03(5.22E+02)** |
> |      **F5**      |    5.41E+03(5.41E+03)    | **1.43E+03(7.16E+01)** |   1.49E+03(6.53E+02)   |
> |      **F6**      |    3.00E+05(3.00E+05)    |   8.20E+05(1.09E+05)   | **2.66E+05(3.11E+04)** |
> |      **F7**      |  **1.19E+03(1.19E+03)**  |   8.06E+03(3.59E+02)   |   1.34E+03(2.63E+01)   |
> |      **F8**      |  **8.37E+05(8.37E+05)**  |   2.03E+07(3.52E+05)   |   1.79E+06(2.36E+05)   |
> |      **F9**      |  **3.24E+03(3.24E+03)**  |   1.76E+07(1.87E+06)   |   5.02E+05(3.04E+04)   |
> |     **F10**      |    4.03E+06(4.03E+06)    |   5.75E+07(5.25E+06)   | **3.51E+06(4.43E+05)** |
> |     **F11**      |  **4.28E+02(4.28E+02)**  |   6.48E+03(2.62E+02)   |   5.80E+02(1.90E+02)   |
> |     **F12**      |  **1.24E+09(1.24E+09)**  |   1.74E+10(1.02E+09)   |   2.28E+09(1.34E+08)   |
> |     **F13**      |  **1.12E+03(1.12E+03)**  |   2.48E+03(1.37E+02)   |   1.42E+03(4.37E+01)   |
> |     **F14**      |  **1.02E+01(1.02E+01)**  |   1.14E+02(1.16E+01)   |   1.64E+01(9.05E-01)   |
> |     **F15**      |  **4.32E+03(4.32E+03)**  |   9.31E+03(1.84E+02)   |   4.75E+03(3.34E+02)   |
> |     **F16**      |  **4.91E+01(4.91E+01)**  |   6.70E+01(1.73E+00)   |   5.79E+01(2.12E+00)   |
> |     **F17**      |  **2.19E+00(2.19E+00)**  |   1.06E+01(8.56E-01)   |   3.14E+00(1.84E-01)   |
> |     **F18**      |  **8.29E+00(8.29E+00)**  |   3.54E+01(2.41E+00)   |   1.29E+01(7.45E-01)   |
> |     **F19**      |  **7.72E+00(7.72E+00)**  |   1.03E+02(6.85E+00)   |   1.56E+01(8.92E-01)   |
> |     **F20**      | **-4.67E+00(-4.67E+00)** |   3.41E+05(2.53E+04)   |   5.10E+03(7.19E+02)   |
> |     **F21**      |  **7.79E+01(7.79E+01)**  |   8.61E+01(1.45E-01)   | **7.02E+01(3.09E+00)** |
> |     **F22**      |    8.10E+01(8.10E+01)    |   8.46E+01(1.08E+00)   | **7.08E+01(7.65E-01)** |
> |     **F23**      |    1.67E+00(1.67E+00)    |   1.94E+00(1.02E-01)   | **1.64E+00(1.29E-02)** |
> |     **F24**      |    7.34E+03(7.34E+03)    |   1.56E+04(3.21E+02)   | **7.31E+03(4.71E+02)** |
> | **win/tie/loss** |          -/-/-           |         22/1/1         |         14/2/8         |
>
> We hope our reply will be recognized by you.
>
> **weakness 4**
>
> POM does not overfit the training set, and experimental results show that POM exhibits generalization capabilities far beyond its training distribution.
>
>
>
> **weakness 5**
>
> Thank you for your valuable comments. We have added these references in the Related Work section, which makes our related work more complete （see Section 2, LLM for Optimization）.
>
> “
>
> *LLaMoCo [38] and EoH [39] use LLM to generate code to solve optimization problems, but the performance of LLaMoCo depends on carefully designed instructions and prompts, and EoH has expensive evaluation costs.*
>
> ”
>
>
>
> **Question 1**
>
> We think your suggestion is very correct and reasonable. In fact, in all BBOB-related experiments, we performed corresponding projections or rotations on the objective function. We only removed the deviation from the optimal solution in the main experimental part.
>
> **Question 2**
>
> The implementation of these baseline methods is described in detail in Appendix section D, and their parameter settings are described in detail in Appendix section E.
>
> **Question 3**
>
> They were not trained on the same training set. We did not need to train LES and LGA, as they claim to be trained models that can be used out of the box. Second, the training frameworks of LES and LGA are not open source, so we could not train them. Third, the training task sets of LES and LGA contain many BBOB functions, which is beneficial to them. Even so, their performance is not as good as POM.

---

> > ### Comment · Reviewer_Se7V · 2024-08-08
> > **Response to the Authors**
> >
> > Thanks for the response from the authors, the response addressed most of my questions.
> > Two further minor questions are as follows,
> > 1. In the newly provided results on BBOB 500D, data in the columu GLHF seems a bit weird that the optimal solutions and the standard deviation are the same.
> > 2. For the BBOB testsuits, do you mean that you use the default projections and rotations provided in the original problems set?

---

> ### Author Response · Authors · 2024-08-08
>
> **Q1**
>
> We deeply apologize for the oversight in the configuration of our experimental code, which led to an error where the standard deviations were identical in the experimental results provided during the rebuttal period. We have rerun the experiment and obtained the correct mean values (and standard deviations), as shown in the table below. We are pleased to report that the performance of the POM method remains significantly superior to the best baselines, which is consistent with our previous conclusions. We are very grateful for the careful review by the reviewers.
>
> |      **F**       |         **POM**         |       **CMA-ES**       |       **LSHADE**       |
> | :--------------: | :---------------------: | :--------------------: | :--------------------: |
> |      **F1**      | **1.29E+02(2.60E+00)**  |   6.12E+02(3.15E+01)   |   2.06E+02(2.18E+01)   |
> |      **F2**      |   5.75E+00(4.71E-01)    |   6.28E+01(3.25E+00)   | **3.13E+00(5.25E-01)** |
> |      **F3**      |   4.72E+03(4.77E+01)    |   8.93E+03(2.57E+02)   | **4.00E+03(1.15E+02)** |
> |      **F4**      |   1.02E+04(7.66E+02)    |   2.62E+04(3.08E+03)   | **8.82E+03(5.22E+02)** |
> |      **F5**      |   5.46E+03(7.62E+01)    | **1.43E+03(7.16E+01)** |   1.49E+03(6.53E+02)   |
> |      **F6**      |   3.18E+05(3.59E+04)    |   8.20E+05(1.09E+05)   | **2.66E+05(3.11E+04)** |
> |      **F7**      | **1.13E+03(3.76E+01)**  |   8.06E+03(3.59E+02)   |   1.34E+03(2.63E+01)   |
> |      **F8**      | **8.62E+05(2.71E+04)**  |   2.03E+07(3.52E+05)   |   1.79E+06(2.36E+05)   |
> |      **F9**      | **3.24E+03(1.72E-01)**  |   1.76E+07(1.87E+06)   |   5.02E+05(3.04E+04)   |
> |     **F10**      |   3.53E+06(3.87E+05)    |   5.75E+07(5.25E+06)   | **3.51E+06(4.43E+05)** |
> |     **F11**      | **4.25E+02(2.61E+01)**  |   6.48E+03(2.62E+02)   |   5.80E+02(1.90E+02)   |
> |     **F12**      | **1.19E+09(6.29E+06)**  |   1.74E+10(1.02E+09)   |   2.28E+09(1.34E+08)   |
> |     **F13**      | **1.13E+03(1.48E+01)**  |   2.48E+03(1.37E+02)   |   1.42E+03(4.37E+01)   |
> |     **F14**      | **1.02E+01(2.52E-01)**  |   1.14E+02(1.16E+01)   |   1.64E+01(9.05E-01)   |
> |     **F15**      | **4.15E+03(9.83E+01)**  |   9.31E+03(1.84E+02)   |   4.75E+03(3.34E+02)   |
> |     **F16**      | **4.67E+01(1.61E+00)**  |   6.70E+01(1.73E+00)   |   5.79E+01(2.12E+00)   |
> |     **F17**      | **2.18E+00(2.41E-02)**  |   1.06E+01(8.56E-01)   |   3.14E+00(1.84E-01)   |
> |     **F18**      | **8.40E+00(8.93E-02)**  |   3.54E+01(2.41E+00)   |   1.29E+01(7.45E-01)   |
> |     **F19**      | **7.62E+00(3.10E-01)**  |   1.03E+02(6.85E+00)   |   1.56E+01(8.92E-01)   |
> |     **F20**      | **-3.43E+00(3.29E+00)** |   3.41E+05(2.53E+04)   |   5.10E+03(7.19E+02)   |
> |     **F21**      |   7.76E+01(2.88E-01)    |   8.61E+01(1.45E-01)   | **7.02E+01(3.09E+00)** |
> |     **F22**      |   8.07E+01(3.29E-01)    |   8.46E+01(1.08E+00)   | **7.08E+01(7.65E-01)** |
> |     **F23**      | **1.60E+00(7.20E-02)**  |   1.94E+00(1.02E-01)   |   1.64E+00(1.29E-02)   |
> |     **F24**      | **7.31E+03(2.04E+02)**  |   1.56E+04(3.21E+02)   |   7.31E+03(4.71E+02)   |
> | **win/tie/loss** |          -/-/-          |         22/1/1         |         14/3/7         |
>
> **Q2**
>
> We did not utilize the default projections and rotations provided in the original problem set because we aimed to test the model's performance on a broader range of scenarios on the BBOB benchmark. Consequently, we strictly adhered to the standard BBOB procedure to generate a series of BBOB function parameters, including projections and rotations.

---

> > ### Comment · Reviewer_Se7V · 2024-08-08
> > **Response to the Authors**
> >
> > Thanks the authors for replying, all of my concern has been addressed. Good work and I would like to raise my score to 7.

---

> > > ### Author Response · Authors · 2024-08-09
> > >
> > > Thank you immensely for your valuable recognition and support towards our work. Your insightful suggestions have not only guided us but also significantly contributed to the enhancement of our paper. We sincerely appreciate your time and effort in helping us refine our research.

---

### Official Review · Reviewer_eUpS · 2024-07-12

**Soundness:** 2
**Presentation:** 1
**Contribution:** 2
**Rating:** 5
**Confidence:** 1

**Summary:**

This paper studies zero-shot optimizers for blackbox optimization problem. The core idea is to pretrain a hypernetwork that generate suitable optimization strategies on a subset of tasks; at test time, the hypernetwork can thus be deployed to propose the optimizer for a given unseen task. The key technical contribution of this work includes the architectural design for this hypernetwork as well as a meta learning algorithm for training it. Empirical studies are conducted on BBOB benchmark as well as Bipedal Walker and Enduro dataset, where the proposed method achieves superior final results compared with several prior arts.

**Strengths:**

1. The idea is straightforward yet grounded in the rich existing literature of hypernetwork in AutoML.
2. The hypernetwork, trained in a meta-learning fashion, exhibits fairly consistent generalization ability on the task considered.
3. Components of the proposed method are extensively studied.

**Weaknesses:**

1. Presentation: Some parts of this paper are a bit vanilla and takes some efforts to follow. For example, section 3 jumps right into detailing the architectural design. Providing high level overview as well as prioritizing the most important designs could benefit the readability of this work.
2. Analysis. The empirical results suggest that the proposed method performs strongly on high dimensional scenarios, but it seems the paper did not provide the intuition behind why it is the case.

**Questions:**

On Bipedal Walker task, the proposed method seems to exhibit substantially higher variance compared with other baselines. Are there any intuitions for this behavior?

**Limitations:**

I did not seem to find any discussion on the limitations.

---

> ### Author Rebuttal · Authors · 2024-08-07
>
> **weakness 1**
>
> Thank you very much for your valuable suggestions. We think your suggestion can really help us enhance the readability of the paper. We have made the following changes to Section 3: 1) First, we briefly introduced the overall model architecture of POM and gave the overall model structure diagram; 2) Further, we introduced the core components of POM in detail. 3) Finally, we introduced the details of POM training and testing.
>
> I hope our changes will be recognized by you!
>
>
>
> **weakness 2**
>
> In the paper, we have made a visual analysis of why POM achieves such excellent performance (see section 4.4). We found that both LMM and LCM can adaptively adjust their own optimization strategies to achieve a good balance between exploration and utilization. Specifically, in the LMM module, we found the following phenomenon (lines 284-289):
>
> *“ 1) Generally, superior individuals receive higher weights during LMM, showcasing POM's ability to balance exploration and exploitation as the population converges. 2) Across diverse function problems, POM dynamically generates optimization strategies, highlighting its adaptability and contributing to robust generalization. 3) Disadvantaged individuals exhibit a more uniform weight distribution, potentially aiding in their escape from local optima and enhancing algorithm convergence.”*
>
> In the LCM module, we found the following phenomenon (see lines 291-299):
>
> *“LCM displays the capacity to adaptively generate diverse strategies for individuals across different ranks in the population, revealing distinct patterns among tasks and rankings. Notably, top-ranking individuals within the top 20, such as those ranked 1st, 5th, and 18th, exhibit a flexible crossover strategy. The dynamic adjustment of crossover probability with population evolution aids in preserving dominant genes and facilitating escape from local optima. Conversely, lower-ranking individuals show an increasing overall probability of crossover, promoting exploration of disadvantaged individuals and enhancing the algorithm's exploration capability. LCM proficiently generates adaptive crossover strategies across tasks, individuals, and convergence stages, significantly boosting both convergence and exploration capabilities.”*
>
> POM has obtained strong optimization and generalization capabilities through pre-training, and has a very strong optimization efficiency. The performance of POM on all problems far exceeds the baselines of other manually designed optimization strategies (DE, ES, LSHADE, CMAES) and the baselines that obtain optimization capabilities through pre-training (LGA, LES), which proves the effectiveness and rationality of POM.
>
>
>
> **Question 1**
>
> The Bipedal Walker task is a very difficult robot control task with very sparse reward signals. This task is very challenging. We can find that LES almost directly failed on this task, while DE, ES, LGA and LSHADE all showed premature convergence. These poor-performing algorithms all exhibit a common characteristic: small variance. Both POM and CMAES, which perform best on this task, exhibit significantly higher variance than other algorithms, which implies that stronger exploration capabilities are required to solve this task. For example, when CMA-ES encounters an evolutionary failure, it will increase the standard deviation of the search distribution and explore better points from a larger space; POM will adaptively adjust the optimization strategy. Among them, the mask module enhances the randomness of POM and avoids the LMM module's over-reliance on outstanding individuals in the population, thus improving the exploration ability of POM.
>
> We did observe in the experiment that there are a large number of failed evolutions in the population evolution process. It may be that after many consecutive generations of attempts, a new and better individual will suddenly be found. This results in a strong randomness in the convergence process, so the convergence curves of CMA-ES and POM appear to have relatively large variances.
>
> **limitations**
>
> Thank you very much for your comments! We have conducted a very detailed analysis of POM in the paper. We also pointed out the advantages and limitations of POM that we observed from the experiments. We have added a summary of limitations as follows:
>
> *"limitations*
>
> 1) *Model size: In the experiment, we found that the relationship between the model size and the performance of POM is not a strict linear relationship. Although the larger the model, the more difficult it is to train, there is still no very quantitative design criterion between model size, training data volume and training difficulty.*
> 2) *Time performance: We introduced an operation similar to the attention mechanism, whose time complexity is $O(n^2)$, which makes POM require a lot of time cost when processing large-scale populations. How to reduce and improve the time efficiency of POM is also worthy of further study.*"

---

> ### Author Response · Authors · 2024-08-12
>
> We hope that our responses have adequately addressed the concerns you previously raised. Your further insights would be greatly appreciated, as they are instrumental in enhancing the quality of our work. We understand that you are busy, and we truly value the time and effort you put into this process. Thank you in advance for your continued support.

---

### Official Review · Reviewer_G2ar · 2024-07-12

**Soundness:** 3
**Presentation:** 2
**Contribution:** 3
**Rating:** 6
**Confidence:** 4

**Summary:**

The paper introduces POM, a neural-network-based evolutionary algorithm for black-box optimization. POM is trained on diverse optimization tasks to enable adaptation to new tasks. POM outperforms the baselines on BBOB benchmark and two robot control tasks.

**Strengths:**

The proposed method performs better than the baselines on the BBOB benchmark and two robotic tasks.

**Increased score from 3 to 6 after discussion**

**Weaknesses:**

- The paper is missing some key references [1, 2, 3]. These methods learn from diverse tasks and are able to adapt to new tasks without any finetuning. This invalidates some of the claims made in the paper (lines 28-29, line 78)
- Section 3 is difficult to read without a background section on population-based optimization. It should describe a general evolutionary algorithm first and how POM parameterizes different components of the algorithm with a neural network.
- Section 3.2 is overly mathy with unnecessary equations. This distracts away from the main idea of the proposed model, which is using neural networks to parameterize evolution. For example, equations (6) and (10) are simply describing multi-head attention and feed-forward layers, which are very standard in deep learning and it's unnecessary to present the equations here. It is also redundant to specify each element of the network parameters $\theta_1$ and $\theta_2$, etc.
- Section 3.3 sounds difficult to believe. How can we expect the same set of parameters ($\theta_1$ and $\theta_2$) to learn from many tasks without feeding metadata of each task? For example, if two tasks have conflicting gradient signals, they get canceled out, and the model basically learns nothing. Specifically, for a certain $x_t$, for task $i$, the model should produce $x_{t+1}$, but for task $j$, the model should produce $x_{t+1}'$, and $x_{t+1}$ and $x_{t+1}'$ are conflicting, then how does the model learn in this case?
- Section 4 is confusing without any experiment setup information. Where are the zero-shot and few-shot settings mentioned in the introduction? How is POM adapted to a new objective after training?
- POM has advantages over the baselines because it was pretrained while the baselines were not, so the better performance of POM is not surprising.

[1] Nguyen, Tung, and Aditya Grover. "Transformer Neural Processes: Uncertainty-Aware Meta Learning Via Sequence Modeling." International Conference on Machine Learning. PMLR, 2022.

[2] Nguyen, Tung, Sudhanshu Agrawal, and Aditya Grover. "ExPT: synthetic pretraining for few-shot experimental design." Advances in Neural Information Processing Systems 36 (2024).

[3] Nguyen, Tung, and Aditya Grover. "LICO: Large Language Models for In-Context Molecular Optimization." arXiv preprint arXiv:2406.18851 (2024).

**Questions:**

- What do the authors mean by "zero-shot" optimization? Does it mean optimizing a new objective function without seeing any (x, y) pair from that function (is this even possible)? Or do the authors mean optimizing a new objective function without any gradient training/fine-tuning? If the latter is the case, then there already exists methods that can perform optimization without training/fine-tuning on the new objective (see my comments above).
- How did the author decide on the input to LMM (H) and LCM (Z)?

**Limitations:**

See above.

---

> ### Author Rebuttal · Authors · 2024-08-07
>
> We've carefully reviewed your feedback, addressed all queries, added the refs, and made revisions as per your suggestions. Your further review and acceptance of the updated manuscript would be highly valued.
>
> **Weaknesses 1**
>
> We have included references to [1-3] in the related work section of the POM due to their significant contributions.
>
> *"TNPs [40], ExPT [41] and LICO78 [ 42] use transformer to solve BBO problems and have achieved good results. TNPs requires contextual information of the target problem, and neither ExPT nor LICO can be directly used to solve tasks with different dimensions from the training task."*
>
> Howerver, we respectfully disagree with your viewpoint for the following reasons:
>
> In [1], it is mentioned: "*the number of evaluation points $N$ and the number of context points $m$ are generated from the same uniform distribution as in training.*" And there are three experimental tasks in [1]. When solving these tasks, [1] is retrained separately. These might imply that [1] may not be suitable for addressing Zero-shot Optimization problems.
>
> In Section 2.1 of [2], it is mentioned that “during the adaptation phase, one can use the pretrained model to optimize any objective function $f$ in the same domain $X \subseteq \mathbb{R}^d$ .” In section 4.3 of [3], it is mentioned that “after training, a single LICO model can be used for optimizing various objective functions within the domain $X \subseteq \mathbb{R}^d$ .”  This could suggest that references [2-3] may not be directly applicable to our particular context.
>
> POM overcomes these limitations.
>
>
>
> At the same time, we found these related papers, which also gave us a lot of inspiration.
>
> - Probing the Decision Boundaries of In-context Learning in Large Language Models.
> - ClimaX: A foundation model for weather and climate.
> - Temporal predictive coding for model-based planning in latent space.
>
>
>
> **Weaknesses 2**
>
> Appendix A provides an overview of the background knowledge related to population-based optimization. Owing to constraints on space, we have chosen to include this information in the appendix rather than the main text. Section 3 meticulously outlines the specifics of POM, thereby ensuring that the reproduction of POM is feasible without a priori knowledge of population-based optimization techniques.
>
> **Weaknesses 3**
>
> Sec. 3.2 details POM's structure and mechanism for reproducibility. Eq. (6) and (10) present specialized model designs, not standard multi-head attention or feed-forward layers, including weights, activations, and normalization.
>
>
>
> **Weaknesses 4**
>
> POM does not have the conflict you claim. The usage of POM is similar to CMA-ES. It can directly optimize the target task without any metadata. POM learns general optimization strategies instead of fitting a specific optimization task. It only needs to input the fitness of the population and some features within the population to adaptively generate optimization strategies.
>
> **Weaknesses 5**
>
> We introduced the concepts of Zero-shot Optimization and Few-shot Optimization in section 3.1 (L87-L90).
>
> The experiment in **Section 4.2** verified the performance of POM in the Zero-shot Optimization scenario (see Fig. 2, 3 and the corresponding appendix for the results). In this section, we directly used the trained POM to solve the target problem.
>
> In **Section 4.3 Fine-tuning Test**, we further verified the performance of POM in the **Few-shot Optimization** scenario. Here, we fine-tune POM using a small number of simulated functions of the target task. The experimental results are shown in Fig. 6.
>
> The trained POM can be directly used to handle new target tasks, which is described in detail in Algorithm 2 (line 167).
>
>
> **Weaknesses 6**
>
> This is the advantage of POM. The baselines include two algorithms (LES and LGA) pre-trained by Meta-BBO. Although they have also been pre-trained, their performance are far inferior to POM.  POM can obtain superior performance through pre-training, while baselines cannot.
>
> **Question 1**
>
> The concept of Zero-shot Optimization introduced in section 3.1 is as follows:
>
> “**Definition 1 Zero-shot Optimization**. An optimizer is applied directly to solve $f$ without any tuning.”
>
> The references [1-3] mentioned in Q1 cannot achieve zero-shot optimization. For details, see the reply to Q1.
>
> **Question 2**
>
> The details of these designs are described in detail in section 3.2.
>
> **LMM**: The role of LMM is to generate a set of candidate solutions $\mathbf{V}^t$ based on the population $\mathbf{X}^t$. Therefore, the input of LMM is based on fitness information, that is, $\mathbf{H}^t$ (see lines 107-118 for details). The input information of LMM does not include the solution $\mathbf{x}$, because $\mathbf{x}$ will have different dimensions and follow different distributions under different tasks. If the input information includes $\mathbf{x}$, POM will not be able to generalize to different task dimensions.
>
> **LCM**:  The input of LCM needs to take into account the fitness information $[\hat{f}_i^t,\hat{r}^t_i]$ of  $\mathbf{x}^t_i$. Intuitively,  if $\mathbf{x}^t_i$ has a low fitness and a poor ranking in the population, then its genes are likely to be eliminated and should not enter the offspring. At the same time, LCM should take into account the cosine similarity between $\mathbf{x}^t_i$ and its candidate solutions $\mathbf{v}^t_i$. Intuitively, if the similarity between $\mathbf{x}^t_i$ and its candidate solution $\mathbf{v}^t_i$ is very low, selecting more genes of the candidate solution $\mathbf{v}^t_i$ into the offspring individuals helps encourage the exploration of the model.

---

> > ### Comment · Reviewer_G2ar · 2024-08-12
> >
> > I thank the authors for the response.
> >
> > I still believe the writing can be significantly improved for better clarity. The main text should provide sufficient background and motivate the approach well before delving into the technical details. Moreover, the methodology section is a bit too verbal and mathy, and some equations are unnecessary. I do not see a difference between Eq (6) and (10) and the standard feed forward and attention layers.
> >
> > Regarding the baselines, TNP, ExPT, and LICO can all be used for zero-shot optimization because they do not require further fine-tuning after pretraining. However, they do require the new objective to lie in the same dimension/domain as the pretraining tasks. I do not think the other papers the authors mentioned are relevant to this work.
> >
> > Can the authors explain why POM does not have the conflicting issue I mentioned in the original review? Imagine there are two tasks in pretraining that have opposite objectives, I do not see how POM can learn a general optimization strategy from both tasks that have conflicting gradient information without any contextual information from each task.
> >
> > I will keep my original score for now.

---

> ### Author Response · Authors · 2024-08-12
>
> We hope that our responses have adequately addressed the concerns you previously raised. Your further insights would be greatly appreciated, as they are instrumental in enhancing the quality of our work. We understand that you are busy, and we truly value the time and effort you put into this process. Thank you in advance for your continued support.

---

> ### Author Response · Authors · 2024-08-13
> **Reply to new discussion (1/3)**
>
> First and foremost, I would like to express my sincere gratitude for your comments and suggestions regarding our manuscript submitted to NeurIPS 2024.  We have diligently revised our paper in accordance with your feedback.
>
> We earnestly hope that you will reconsider our revised manuscript and appreciate the improvements made. If you find that our efforts and the results now meet the standards of the conference, we kindly request that you consider increasing the score of our paper. We highly value your assessment and look forward to your final approval.
>
> Thank you once again for your valuable time and expertise.
>
> ## Q1: I still believe the writing can be significantly improved for...
>
> Thank you for your valuable suggestions, which have helped us improve the expression of our paper. To better introduce the background knowledge and clarify the motivation behind the design of POM, we have followed your advice and added the following content in Section 3 before detailing the mechanism of POM (see Section 3.2 for details).
>
> "
> ***3.2 Classic Population Optimization Algorithm***
> *In this section, we use Differential Evolution (DE) as an example to review classic evolutionary algorithms. DE [20,43] is a prominent family within evolutionary algorithms (EAs), known for its advantageous properties such as rapid convergence and robust performance [44,45]. The optimization strategy of DE primarily involves mutation and crossover operations.*
>
> *The classic DE/rand/1 crossover operator is illustrated in Eq. (1) (additional examples are listed in Appendix A.2). Each mutation strategy can be viewed as a specific instance of Eq. (2); Further details are provided in Appendix A.2. Additionally, we represent the mutation strategy in a matrix form, as shown in Eq. (3). The matrix $\mathbf{S}$ evolves with the generation index *t*, indicating that the mutation strategy adapts across different generations. Consequently, we propose a module to enhance the performance of the mutation operation, which leverages the information from the population of the *t*th generation to generate $\mathbf{S}^t$. This serves as the motivation for our design of the LMM.*
>
> $\mathbf{v}\_i^t = \mathbf{x}\_{r1}^t+F\cdot(\mathbf{x}\_{r2}^t-\mathbf{x}\_{r3}^t)$  &nbsp;&nbsp;&nbsp; (1)
>
> *In the crossover phase at step *t*, DE uses a fixed crossover probability $cr_i^t$ ∈ [0,1] for each individual $\mathbf{x}_i^t$ in the population, as shown in Eq. (9). The crossover strategy for the entire population can then be expressed as a vector $\mathbf{cr}^t = (cr_1^t, cr_2^t, ..., cr_N^t)$. Our goal is to design a module that adaptively generates $\mathbf{cr}^t$ using the information from the population. This approach allows for the automatic design of the crossover strategy by controlling the parameter *cr*. This serves as the motivation for our design of LCM.*

---

> ### Author Response · Authors · 2024-08-13
> **Reply to new discussion (2/3)**
>
> ## Q2: Moreover, the methodology section is a bit too verbal and mathy, and some equations are unnecessary. I do not see a difference between Eq (6) and (10) and the standard feed forward and attention layers.
>
> In this section, we present detailed formulas to ensure the reproducibility of POM, which are also crucial for elucidating the mechanism of POM. We will then discuss in detail the differences between these formulas and the standard self-attention and feed-forward layers.
>
> ### The difference between Eq (6) and standard self-attention
>
> **A. Self-Attention**
>
> 1. **Input Transformation**:
>
>    $$ \mathbf{Q}= \mathbf{X}\mathbf{W}^Q, \quad \mathbf{K} = \mathbf{X}\mathbf{W}^K, \quad \mathbf{V} = \mathbf{X}\mathbf{W}^V$$
>
> 2. **Attention**:
>
>    $$\text{Self-Attention}(\mathbf{Q}, \mathbf{K}, \mathbf{V}) = \text{softmax}\left(\frac{\mathbf{Q}\mathbf{K}^T}{\sqrt{d_k}}\right)\mathbf{V}$$
>
> 3. **Output**:
>
>    $$\text{Output} = \text{Self-Attention}(\mathbf{Q},\mathbf{K},\mathbf{V})$$
>
> **B. Eq. (6)**
> 1. **Input Transformation**:
> $$
> \mathbf{\hat{H}}^t=\text{Tanh}(\mathbf{H}^t\mathbf{W}_{m1}+\mathbf{b}\_{m1}),\quad \mathbf{Q}^t=Tanh(\mathbf{\hat{H}}^t\mathbf{W}\_{m2}+\mathbf{b}\_{m2}), \quad \mathbf{K}^t=Tanh(\mathbf{\hat{H}}^t\mathbf{W}\_{m3}+\mathbf{b}\_{m3})
> $$
> 2. **Attention**:
>
>    $$\text{LMM-Attention}(\mathbf{Q}, \mathbf{K}, \mathbf{X})= \text{mask}\left(\text{softmax}\left( \text{Tanh}\left(\frac{\mathbf{Q}\mathbf{K}^T}{\sqrt{d_k}}\right)\right),r_{\text{mask}}\right)\mathbf{X}$$
>
> 3. **Output**:
>
>    $$\text{Output} = \text{LMM-Attention}(\mathbf{Q}, \mathbf{K}, \mathbf{X})$$
>
> The differences between Eq (6) and standard self-attention are evident as follows:
>
> 1. **No Calculation of V**: In Eq (6), we do not compute $\mathbf{V}$; instead, we use $\mathbf{X}$ in the attention calculation. This approach is crucial for achieving good performance in our optimization task, whereas standard self-attention struggles to adapt to our problem scenario.
>
> 2. **Different Processing of Input Information**: We introduce the Tanh activation function and a bias term, which helps enhance the performance of POM. This is a significant departure from the standard self-attention mechanism.
>
> 3. **Attention Matrix Mapping and Masking**: For the computed attention matrix, we apply the Tanh function to map it to the range $[-1,1]$. Additionally, we use a random mask operation, setting elements to zero with a probability of $r_{mask}$.
>
> In summary, there are several key differences between our approach and standard self-attention. These differences are crucial for the performance of POM, and it is essential to detail them thoroughly. This is not a shortcoming but rather a necessary distinction.
>
> ### 2 Eq (10) and feed forward layer
> **A. feed forward**
> $$\text{Output} = \mathbf{W}_2 \cdot \text{ReLU}(\mathbf{W}_1 \cdot \mathbf{X} + \mathbf{b}_1) + \mathbf{b}_2$$
>
> **B. Eq (10)**
> $$\mathbf{cr}^t = \text{sigmoid}\left(\text{layernorm}\left(\tanh\left(\mathbf{Z}^t \times \mathbf{W_{c1}} + \mathbf{b_{c1}}\right) | \mathbf{\tau}\right) \times \mathbf{W_{c2}} + \mathbf{b_{c2}}\right)$$
>
>
> It is worth noting that, although the two are very similar in form, Eq (10) provides a detailed implementation scheme. Any modifications to Eq (10), such as replacing the sigmoid function with ReLU, result in a significant decline in POM's performance. This is because we rely on the sigmoid function to map the output to the range $[0,1]$, representing probability values. Changes to other activation functions or hyperparameters in Eq (10) also lead to performance degradation.
>
> ## Q3: Regarding the baselines, TNP, ExPT, and LICO can all be used for zero-shot optimization because they do not require further fine-tuning after pretraining. However, they do require the new objective to lie in the same dimension/domain as the pretraining tasks.
>
> It is clear that TNP, ExPT, and LICO require retraining in different dimensional scenarios when faced with tasks of varying dimensions. This does not align with the definition of Zero-shot Optimization, which is "an optimizer that can be directly applied to solve a continuous black-box optimization problem $f$ without any tuning."
>
>
> ***"Definition 1 Zero-shot Optimization** : Zero-shot optimization refers to an optimizer that is applied directly to solve a continuous black-box optimization problem $f$ without any tuning. This means that the optimizer does not require any contextual information about
> $f$ and can be directly used to handle problems of any dimensionality."*

---

> ### Author Response · Authors · 2024-08-13
> **Reply to new discussion (3/3)**
>
> ## Q4: Can the authors explain why POM does not have the conflicting issue I mentioned in the original review...
>
>  Conflicts between optimization objectives do not necessarily lead to conflicts in optimization strategies. POM focuses on learning the mapping from the ranking information of individuals within a population to the search strategies, rather than adopting a fixed optimization strategy to suit all tasks.
>
> Specifically, the LMM module adaptively generates the strategy matrix $\mathbf{S}^t$ using the population's ranking information ($\hat{r}^t_i$) and the information representing their relative advantages $\hat{f}_i^t$ (normalized from $f(\mathbf{x}_i^t)$, representing individual $i$. Note that we can convert all problems to minimization problems by adding a negative sign).
>
> The LCM module uses the population's ranking information ($\hat{r}^t_i$), the relative advantage information of individuals within the population $\hat{f}_i^t$, and the similarity information between parent and offspring $sim_i^t$ to adaptively generate the crossover strategy.
>
> This means that POM is not fitting a specific problem $f$, nor is it searching for an optimal $\mathbf{x}^* \in \mathbb{R}^d$ in its solution space. Instead, it searches for optimal POM strategy parameters $\mathbf{\theta}^*$ in the strategy space through these training tasks. Therefore, even if there are conflicting objective functions, it does not affect the performance of POM. Furthermore, the training algorithm proposed by POM, MetaGBT, integrates gradient information from a set of tasks, making the training process more stable and easier.
>
> Let's simulate the situation you mentioned. Suppose there is a function $f(\mathbf{x})=\sum_i{x_i}$, with task one being $f_1=\mathop{\arg\max}\limits_{\mathbf{x}}f(\mathbf{x})$ and task two being $f_2=\mathop{\arg\min}\limits_{\mathbf{x}}f(\mathbf{x})$. Clearly, these two tasks are in direct conflict.
>
> Assuming that at the current moment, the populations for both tasks are the same, they are in the following state:
>
> $$
> \mathbf{X}=
> \begin{bmatrix}
> 1 & 2 & 3 \\\\
> 4 & 5 & 6 \\\\
> 7 & 8 & 9 \\\\
> 1 & 3 & 4 \\\\
> \end{bmatrix}
> \mathbf{X}_1=
> \begin{bmatrix}
> 7 & 8 & 9 \\\\
> 4 & 5 & 6 \\\\
> 1 & 3 & 4 \\\\
> 1 & 2 & 3 \\\\
> \end{bmatrix}
> \mathbf{X}_2=
> \begin{bmatrix}
> 1 & 2 & 3 \\\\
> 1 & 3 & 4 \\\\
> 4 & 5 & 6 \\\\
> 7 & 8 & 9 \\\\
> \end{bmatrix}
> $$
>
> $$
> \mathbf{U}_1=\mathbf{CR}\cdot\mathbf{V}_1+(1-\mathbf{CR})\cdot\mathbf{X}_1=
>  \left[
>  \begin{matrix}
> 6.96 & 7.96 & 8.96 \\\\
> 4.54 & 5.55 & 6.55 \\\\
> 3.25 & 4.79 & 5.79 \\\\
> 4.51 & 5.65 & 6.64 \\\\
>   \end{matrix}
>   \right]
> $$
>
> $$
> \mathbf{U}_2=\mathbf{CR}\cdot\mathbf{V}_2+(1-\mathbf{CR})\cdot\mathbf{X}_2=
>  \left[
>  \begin{matrix}
> 1.02 & 2.03 & 3.03 \\\\
> 1.14 & 2.89 & 3.89 \\\\
> 2.68 & 3.83 & 4.83 \\\\
> 2.27 & 3.54 & 4.54 \\\\
>   \end{matrix}
>   \right]
> $$
>
> Finally, after performing selection between $\mathbf{X}$ and $\mathbf{U}$, the resulting population is:
> $$
> \mathbf{X}_1'=
>  \left[
>  \begin{matrix}
> 7.00 & 8.00 & 9.00 \\\\
> 4.54 & 5.55 & 6.55 \\\\
> 3.25 & 4.79 & 5.79 \\\\
> 4.51 & 5.65 & 6.64 \\\\
>   \end{matrix}
>   \right] \mathbf{X}_2'=
>  \left[
>  \begin{matrix}
> 1.00 & 2.00 & 3.00 \\\\
> 1.14 & 2.89 & 3.89 \\\\
> 2.68 & 3.83 & 4.83 \\\\
> 2.27 & 3.54 & 4.54 \\\\
>   \end{matrix}
>   \right]
> $$
>
> **The average fitness for Task 1 improved from 13.25 to 17.8175, demonstrating an enhancement in the performance of Task 1. Conversely, the average fitness for the population of Task 2 decreased from 13.25 to 8.9025, which also indicates an improvement in the performance of Task 2.**
>
> **This shows that even with conflicting objective functions, Their optimization strategies do not necessarily conflict. This demonstrates that POM learns a universal optimization strategy, and there is no conflict as you claimed.**

---

> > ### Comment · Reviewer_G2ar · 2024-08-14
> >
> > I thank the authors for the response. Your clarification has addressed my major concern about the conflicting issue. The detail that helped resolve my confusion was that LCM and LLM used the population's ranking and relative performance, which contain information about the task/objective (the same x can be ranked 1st for objective 1 but ranked last for objective 2). I increased my score from 3 to 6.
> >
> > With regard to writing, I was aware of the differences in some design choices of the proposed method with standard layers, but I still believe these details should be deferred to later sections so that readers can appreciate the main idea.

---

> > > ### Author Response · Authors · 2024-08-14
> > >
> > > I wish to express my sincere appreciation for the invaluable assistance and dedication you have provided during the review process of my paper. Your insightful comments and thoughtful suggestions have significantly contributed to the development and refinement of my work. Your efforts have been of immense help in enhancing the quality and clarity of my research. Thank you for your generous support. It is greatly appreciated.

---

### Decision · Program_Chairs · 2024-09-25

**Decision:**

Accept (poster)

**Comment:**

This work considers zero-shot black box optimization with a population-based pretrained optimization model (POM). The reviewers appreciate the design and training process of POM under a meta-learning framework,  and the performance is promising.  It receives positive scores after the rebuttal discussions. AC agrees this is a nice progress on zero-shot black box optimization and recommends acceptance. This work can be further strengthened with an improved presentation, more analysis and justification on its performance especially under high-dimensional scenarios, and generalization studies on more realistic complex cases.